# Virtual brain twins for stimulation in epilepsy

**Huifang E. Wang** [1] ✉, **Borana Dollomaja**[1], **Paul Triebkorn** [1],
**Gian Marco Duma**[1,2], **Adam Williamson**[1], **Julia Makhalova**[3,4,5],
**Jean-Didier Lemarechal**[1], **Fabrice Bartolomei**[1,3] & **Viktor Jirsa** [1] ✉

Estimating the epileptogenic zone network (EZN) is an important part of the diagnosis of drug-resistant focal epilepsy and has a pivotal role in treatment and intervention. Virtual brain twins provide a modeling method for personalized diagnosis and treatment. They integrate patient-specific brain topography with structural connectivity from anatomical neuroimaging such as magnetic resonance imaging, and dynamic activity from functional recordings such as electroencephalography (EEG) and stereo-EEG (SEEG). Seizures show rich spatial and temporal features in functional recordings, which can be exploited to estimate the EZN. Stimulation-induced seizures can provide important and complementary information. Here we consider invasive SEEG stimulation and non-invasive temporal interference stimulation as a complementary approach. This paper offers a high-resolution virtual brain twin framework for EZN diagnosis based on stimulation-induced seizures. It provides an important methodological and conceptual basis to make the transition from invasive to non-invasive diagnosis and treatment of drug-resistant focal epilepsy.

In the most complex cases of drug-resistant focal epilepsy, accurate diagnosis requires invasive stereo-electroencephalography (SEEG) implantation. This procedure is crucial for estimating the epileptogenic zone network (EZN), a key element for successful treatment[1,2]. SEEG has become one of the principal techniques for delineating EZNs[3,4]. In the past 15 years, several data analysis methods for quantifying EZNs have been proposed based on the spectral analysis of SEEG signals[5,6]. Beyond pure data-driven analysis approaches, several methods linking mechanistic models and data analysis have been developed[7–12], formally exploiting causal hypotheses within an inference framework. We developed a workflow for the estimation of a patient's EZN using personalized whole-brain models, called the virtual epileptic patient (VEP)[13–16]. The VEP workflow was evaluated retrospectively using 53 patients with 187 spontaneous seizures and is now being evaluated in an ongoing clinical trial (EPINOV) with 356 prospective patients with epilepsy[14,15]. The virtual brain twin concept was proposed based on the VEP workflow and has been extended to various brain disorders[17]. Virtual brain twins are personalized, generative and adaptive

brain models based on data from an individual's brain for scientific and clinical use. In this study, we introduce a high-resolution virtual brain twin workflow, specifically designed for estimating the EZN using a stimulation paradigm.

SEEG stimulation, involving direct electrical stimulation through SEEG electrodes, can be used to map brain function, as well as to provoke seizures for better EZN diagnosis, especially when spontaneous seizures are not obtained. SEEG-stimulation-induced seizures (at 1 Hz or 50 Hz, usually with pulses of 1 ms at 1–3 mA) are an important tool for localizing the EZN and are also associated with a better post-surgical outcome[18–20]. First, we propose a high-resolution personalized whole-brain model—a virtual brain twin—dedicated to assessing stimulations performed through SEEG electrodes. Then, we evaluate the capacity of our approach to translate from invasive stimulation and recording via SEEG to non-invasive procedures using scalp-EEG recordings and transcranial electrical stimulation techniques, notably temporal interference (TI) stimulation[21]. The recently developed TI stimulation has the capacity of reaching deeper structures than

[1]Aix Marseille Université, INSERM, INS, Institut de Neurosciences des Systèmes, Marseille, France. [2]Scientific Institute, IRCCS E.Medea, Epilepsy and Clinical Neurophysiology Unit, Conegliano, Italy. [3]Epileptology and Clinical Neurophysiology Department, APHM, Timone Hospital, Marseille, France. [4]Aix Marseille Université, CNRS, CRMBM, Marseille, France. [5]APHM, Timone University Hospital, CEMEREM, Marseille, France. ✉e-mail: huyfang.wang@univ-amu.fr; viktor.jirsa@univ-amu.fr

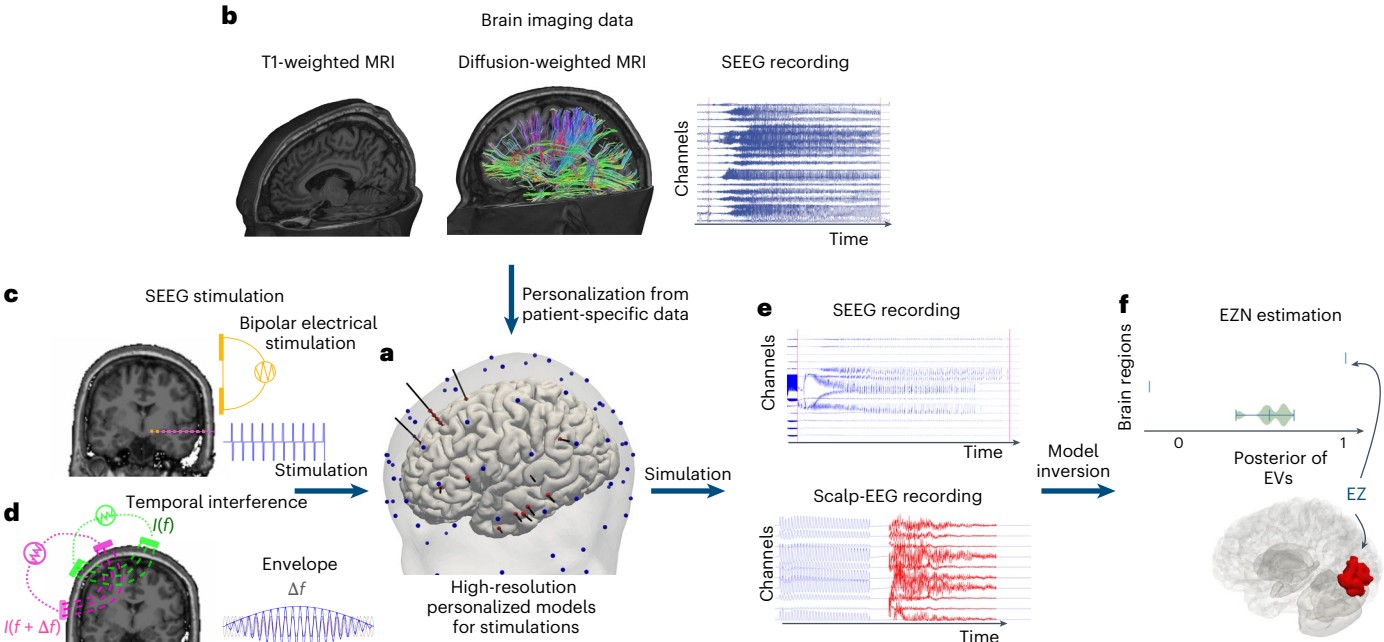

**Fig. 1 | The workflow of the virtual brain twin for estimating the EZN using stimulation techniques. a,b**, A personalized high-resolution model (**a**) is based on individual brain geometry extracted from T1-weighted MRI and structural connectivity from tractography on diffusion-weighted MRI data (**b**). High-resolution virtual brain models simulate neural source activity with spatial resolutions of about 10 mm². The modeling parameters are inferred from the spontaneous SEEG recordings (**b**). **c,d**, We illustrate two types of stimulation: SEEG and TI, to induce seizure activity. **c**, SEEG stimulation uses bipolar stimulation in which two electrodes are used: one serves as the cathode and the other as the anode. The electric current flows between two electrodes, which is parameterized by current amplitude, pulse width and frequency. **d**, TI stimulation applies two current sources (*I*) simultaneously via electrically isolated pairs of scalp electrodes (green and pink) at kliohertz frequencies

*f* and *f* + Δ*f*. The currents generate oscillating electric fields, which results in an envelope amplitude that is modulated periodically at Δ*f*. The electric field influences the brain activity that can be generated by the high-resolution personalized whole-brain model (**a**). The red and blue dots represent SEEG and scalp-EEG electrodes, respectively. **e**, The simulated source activity can be mapped onto the corresponding SEEG and scalp-EEG signals, through the gain matrices, which are constructed based on the locations of SEEG and scalp-EEG electrodes relative to the source vertices. The red curves on the scalp-EEG recordings are plotted using a different scale to visualize the signals following the high-amplitude signals induced by TI stimulation. **f**, By utilizing data features extracted from SEEG and scalp-EEG signals, Bayesian inference methods can estimate a posterior distribution of EVs, suggesting the potential EZN.

conventional transcranial direct- and alternating-current stimulation (tDCS and tACS)[22,23]. Thus, combining the advantages of tDCS and deep brain stimulation[24], TI stimulation is non-invasive, focal and capable of targeting deep brain structures. TI stimulation exploits the brain's insensitivity to high-frequency electric fields in the kilohertz range, as demonstrated in a recent clinical trial with patients with epilepsy[25]. TI stimulation occurs when electric fields generated by multiple electrode pairs with slightly different frequencies interfere at a target location. This interference produces an envelope modulation at a lower frequency—equal to the difference between source frequencies (normally below 150 Hz)—which effectively stimulates the target tissue.

In summary, this study presents a workflow for estimating the EZN using a personalized high-resolution virtual brain twin under a stimulation paradigm. We develop a pipeline that (1) builds a personalized high-resolution brain model for either SEEG or TI stimulation; (2) estimates the EZN from stimulation-induced seizures, and validates it by simulated data; (3) refines the estimation of EZN by integrating multiple recording modalities, such as by combining scalp-EEG and SEEG; and (4) extends the pipeline to incorporate region-specific heterogeneity in local connectivity and support different brain atlases. This research provides a necessary step for (1) a series of scientific and clinical studies, such as optimization of stimulation parameters for diagnosis and treatment; (2) moving from invasive to non-invasive diagnosis and treatment of drug-resistant focal epilepsy; and (3) natural integration of multiple functional data modalities.

## Results

### VEP-stimulation workflow

We built a high-resolution virtual brain twin workflow to estimate the EZN using stimulation techniques. First, we built a virtual brain twin based on available data before stimulation. An EZN is defined as the tissue responsible for generating seizures and may involve distant brain areas characterized by altered excitability[1]. Our model-based high-resolution workflow using perturbation techniques for diagnostic EZN mapping is shown in Fig. 1 and a detailed flowchart is shown in Supplementary Fig. 1. First, high-resolution full-brain network models (Fig. 1a) were established using patient-specific data from patients with epilepsy. The structure of the model was defined by the detailed surface of the cortex and the subcortical volumes from T1-weighted MRI. The global structural connectivity between brain regions through white matter fibers was estimated from diffusion-weighted magnetic resonance imaging (MRI). The cortical surface data were generated from T1-weighted MRI, resulting in surfaces with 20,284 vertices with a vertex area of about 10 mm². We simulated the time series on the patient's specific structural scaffold using the phenomenological Epileptor model[26], a system of differential equations that can describe seizure initiation, propagation and termination, resulting in electrophysiological seizure-like events. The spatial domain of the Epileptor is given by the high-resolution network of neural populations; thus, seizures can propagate locally across neighboring vertices of the cortex and globally through white matter fiber connections[7]. The initial personalized modeling parameters can be inferred from the spontaneous seizure recording[15].

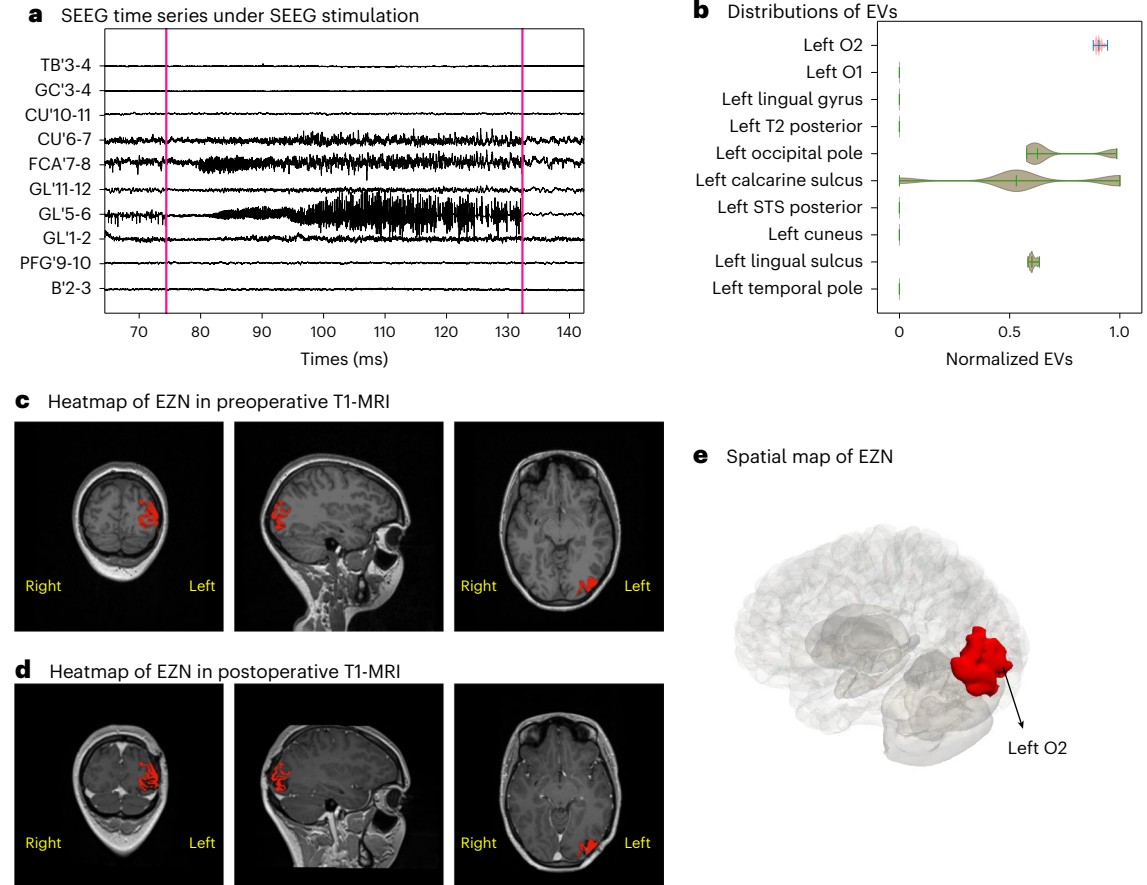

**Fig. 2 | VEP diagnostic mapping for spontaneous seizures (empirical data).** **a**, SEEG recordings from one seizure in a 23-year-old female patient. The left axis shows the names of the selected electrode channels. **b**, Posterior distribution of the EVs (higher value indicates higher probability of seizure) for ten selected regions obtained from the HMC pipeline. Each violin plot shows the distribution of the entire data range using a kernel density estimate. The three bars represent the 25th percentile, the median and the 75th percentile, respectively. All violin plots in this paper follow the same format. **c**, Heatmap of the left O2 identified by VEP (in red) shown in a preoperative T1-MRI. **d**, Heatmap of the left O2 identified by VEP (in red) shown in a postoperative T1-MRI. **e**, The left O2 (in red) was projected on the patient's 3D meshes.

We then developed a high-resolution virtual brain twin designed for stimulation. First, we calculated the electric fields induced by both stimulation methods (Fig. 1c,d). For the SEEG stimulation, two SEEG electrodes serve as cathode and anode to generate bipolar stimulation and electric current flow is parameterized by current amplitude, pulse width and frequency (Fig. 1c). The perturbation effect was applied to the vertices of the high-resolution surface through the SEEG-to-source mapping, with perturbation strength decaying as the distance between vertices and electrodes increased. For TI stimulation, we calculated the amplitude of the envelope of the TI electric field projected onto the surface normal vector at each vertex (Fig. 1d). Both types of stimulation generated an accumulation in the slow state variable $m$ of the Epileptor-stimulation model (equation (1)), which represents stimulation-induced tissue changes. When $m$ reaches a given threshold, the model undergoes a transition into the seizure state. A post-SEEG implantation computed tomography (CT) scan is used to localize SEEG contacts and co-register them with the structural scaffold. Scalp-EEG electrodes were placed on the scalp using the standard international 10-5 system. This high-resolution model allows us to consider detailed electric dipoles, generated by neural activity, for building high-fidelity forward solutions. The source-to-sensor matrix maps the activity from the neural sources—located at vertices of the cortical surface and subcortex—to SEEG or scalp-EEG electrodes, taking into account their orientation and distance.

Model inversion estimates patient-specific brain model parameters, especially epileptogenicity and global network scaling using Hamiltonian Monte Carlo (HMC) sampling techniques from Bayesian inference methods[10,15]. The estimation is based on the structural brain scaffold, modeled seizure dynamics and the data feature extracted from scalp-EEG and/or SEEG seizure recordings (Fig. 1e). The model inversion uses a non-informative prior where all the brain regions have the same prior distribution (the prior assumption is that all regions are healthy). The result is the posterior probability from which the EZN is identified. We also introduced multimodal inference for simultaneous SEEG and scalp-EEG to infer the EZN (Fig. 1f).

## Virtual brain twins for SEEG stimulation

We used the data from a right-handed 23-year-old female patient diagnosed with left occipital lobe epilepsy to illustrate how to use our workflow. We first extracted the brain geometry, the structural connectivity matrix and the source-to-sensor mapping using this patient's T1-weighted MRI, diffusion-weighted MRI and post-SEEG-implantation CT scans. We built the whole-brain neural mass model based on this anatomical information. Then we ran the HMC algorithm for three spontaneous seizures recorded from this patient (one of the three seizures with selected electrodes is shown in Fig. 2a). We pooled the posterior distribution from the three seizures, in which the left lateral occipital cortex (region O2 (in the VEP atlas[27])) was consistently identified as part of the EZN. Then we projected the left O2 onto both preoperative

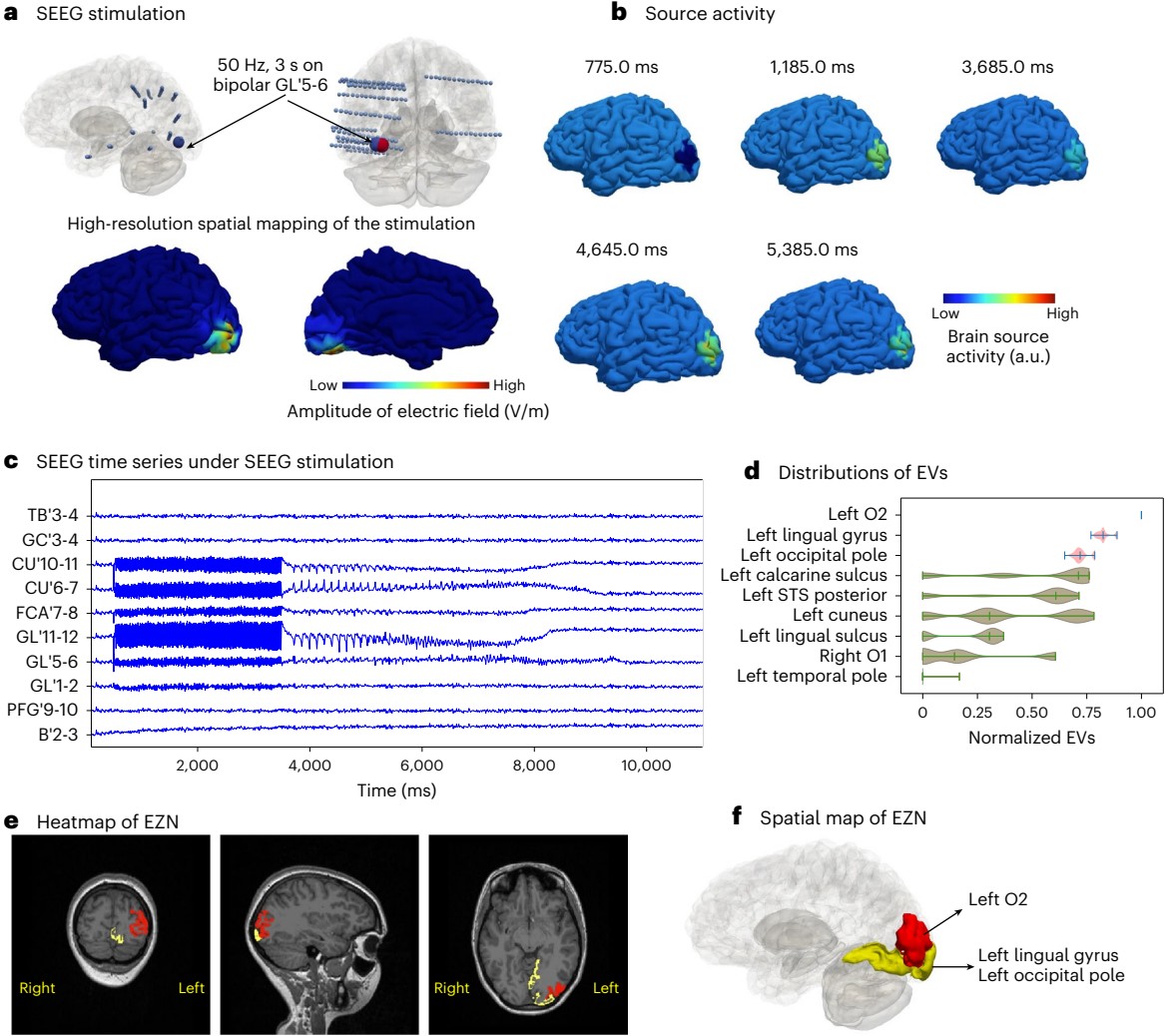

**Fig. 3 | Estimating EZN from a SEEG-stimulation-induced seizure (synthetic data). a**, Top: GL'5-6 (large blue/red sphere) is the stimulated contact in the left occipital lobe, using a bipolar pulse stimulation (50 Hz for a duration of 3.5 s with pulse duration 1 ms). Bottom: spatial map showing the amplitude of electric field (indicated by the color bar) at each brain vertex, induced by SEEG stimulation from contact pair GL'5-6. **b**, Neural source activity in arbitrary units (a.u.) is shown on the cortical mesh at five different time points, with values indicated by color. The seizures are located around the left O2 region of the VEP atlas. **c**, Selected simulated SEEG time series from a SEEG-stimulation-induced seizure.

**d**, Posterior distribution of the EVs (higher value indicates higher chance for seizure) of nine selected regions obtained from HMC sampling. Each violin plot shows the distribution of the entire data range using a kernel density estimate. The three bars represent the 25th percentile, the median and the 75th percentile, respectively. Red regions indicate highest chance of being the EZN; the other areas are in green. **e,f**, The region of the highest EV posterior distribution is the left O2 in red shown in T1-MRI (**e**) and the 3D brain (**f**). The two regions (the left lingual gyrus and the left occipital pole) are shown in yellow.

and postoperative T1-MRI slides and three-dimensional (3D) brain meshes (in Fig. 2c–e). The patient underwent resective surgery, in which a large portion of the left O2 was removed (in Fig. 2d), and was almost seizure-free after surgery (surgery outcome class of Engel II[28]). On the basis of this result, we built the high-resolution whole-brain neural field model for SEEG and TI stimulation to test the hypothesis that the left O2 was the EZN of the patient.

To model direct electrical stimulation by SEEG electrodes in the brain, we first mapped the contribution of SEEG stimulation current to brain source activity. We retrieved the contribution from a pair of electrodes, in which the perturbation is applied to the brain regions based on the sensor-to-source mapping matrix[15]. Then we calculated the effect of the bipolar pulse stimulation (50 Hz for a duration of 3.5 s on bipolar GL'5-6 electrode leads) on each vertex of the cortex mesh (Fig. 3a). We used the Epileptor-Stimulation model at each of the 20,284 vertices. The stimulation leads to an accumulation in a state variable $m$ that pushes the brain to seize (Supplementary Fig. 2). The seizures

were located around the left O2 regions (Fig. 3b and Supplementary Video 1). From these simulated neural source signals, we obtained SEEG signals (Fig. 3c) using the source-to-sensor matrix.

These synthetic SEEG recordings (Fig. 3c) were used for model inversion, to estimate the EZN. We obtained the left O2 with the highest epileptogenic values (EVs) out of all the samples according to the posterior distribution from the HMC algorithm (Fig. 3d). The two neighboring regions (the left lingual gyrus and the left occipital pole) were the second group of candidates that could belong to the EZN. The heatmap projected on the patient's T1-MRI revealed the spatial mapping of the EZN (sagittal, axial and coronal view images shown in Fig. 3e). The spatial mapping of these three regions in 3D brain is highlighted in red and yellow in Fig. 3f.

### Virtual brain twins for TI stimulation
We here demonstrate how the pipeline works for TI stimulation. First, we began with high-resolution twin modeling for this patient based

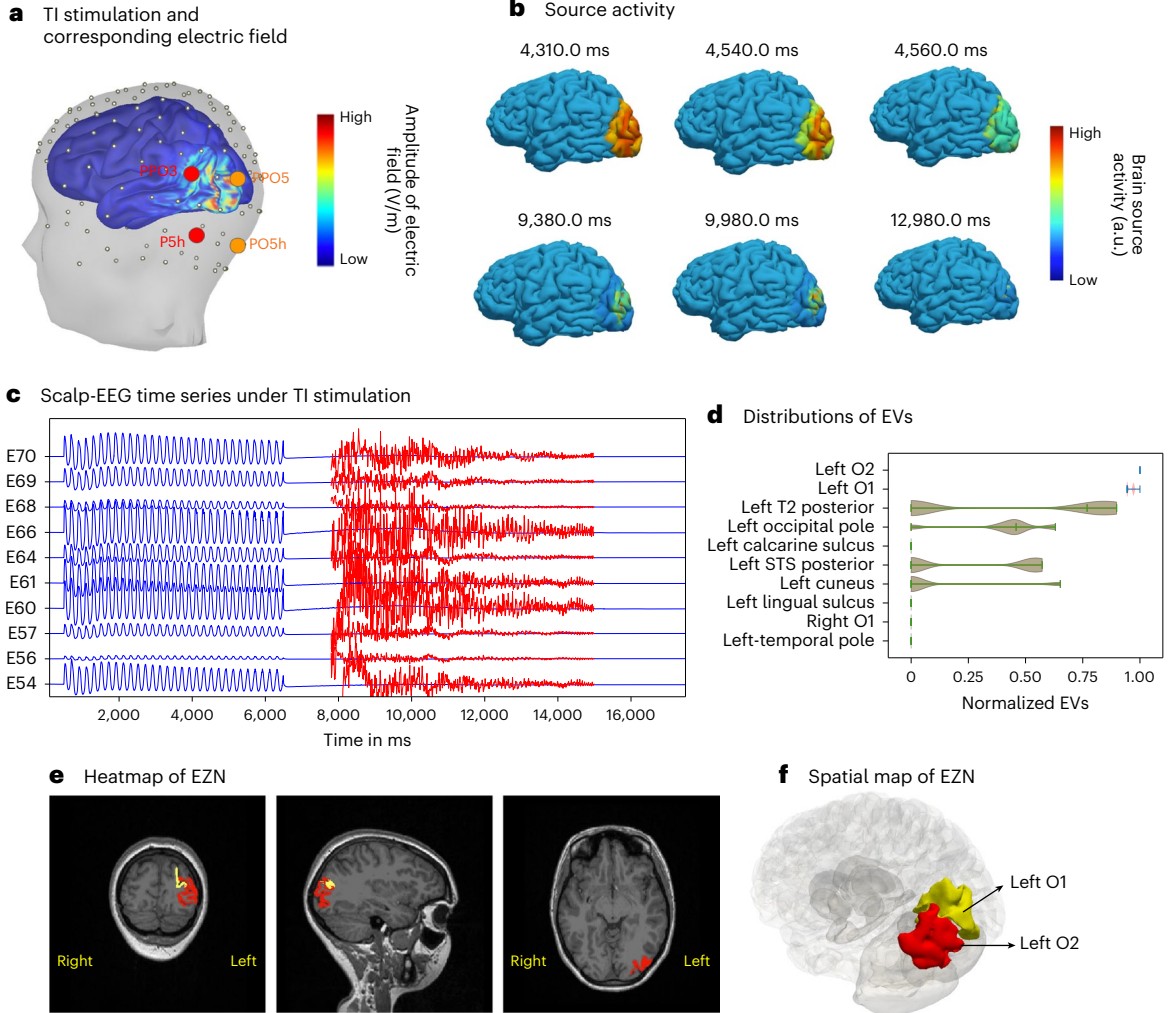

**Fig. 4 | Estimating EZN from TI-stimulation-induced seizure (synthetic data). a**, The electric field of TI stimulation by two pairs of scalp-EEG electrodes (shown in red and orange) based on the 10-5 international reference system, using an extended scalp-EEG cap from SIMNIBS[29]. We applied stimulation at 1,000 Hz and 1,005 Hz through the first (PPO3, P5h) and second (PPO5, PO5h) scalp electrode pairs, respectively. The electrodes PPO3, P5h, PPO5 and PO5h are part of the extended 10−5 EEG system and correspond to intermediate scalp locations over the parietal and parieto-occipital regions. The spatial distribution of the amplitude of the TI electric field is colored in the 3D brain. **b**, Seizure dynamics were simulated using the Epileptor-Stimulation model through the TI stimulation. Neural activity is shown in color on the cortical mesh at six different time points. The seizures are located around the left O2 regions of the VEP atlas. **c**, Simulated scalp-EEG time series from TI-stimulation-induced seizure. The y-axis shows the names of selected scalp-EEG channels. The scaled-up time series during the seizure period is shown in red. **d**, Posterior distribution of the EVs (higher value indicates higher chance for seizure) for ten selected regions obtained from the HMC sampling. Red regions indicate highest chance of being the EZN; the others are in green. **e,f**, The region of the highest EV posterior distribution is the left O2 in red shown in T1-MRI (**e**) and the 3D brain (**f**). The second region (left O1) is shown in yellow.

on spontaneous seizures (Fig. 2). We calculated the TI electric field, which is projected onto the cortical surface by stimulating two pairs of scalp-EEG electrodes (PPO3−P5h and PPO5−PO5h[29] in Fig. 4a) at frequencies of 1,000 Hz and 1,005 Hz, respectively. The stimulation amplitude is determined based on the assumption that the components of the field that have the greatest impact are those that are parallel to the neurons' axons[30]. On the basis of the TI electric field, the stimulation input is applied to the Epileptor model on each vertex of the cortical mesh (Supplementary Fig. 3). We simulated the brain source signals on each vertex of the whole brain by considering both global and local connectivity (Fig. 4b and Supplementary Video 2). We then mapped the activities from the vertices of the cortical surface onto the scalp-EEG signals (Fig. 4c) using the source-to-sensor mapping matrix.

We extracted the data features from these synthetic scalp-EEG recordings (Fig. 4c) as input to the HMC model inversion to estimate the epileptogenic zone. From the posterior distribution of EVs, we obtained the left O2 as first and the left O1 as the second candidate belonging to

the EZN (Fig. 4d). The heatmap of these two regions projected on the patient's T1-MRI shows the spatial mapping of the EZN (sagittal, axial and coronal view images shown in Fig. 4e) and the 3D brain in Fig. 4f, where the left O2 is in red and the left O1 in yellow.

## Multimodal inference

For patients with multiple seizure recordings or simultaneous multiple modality recordings (here we used SEEG and scalp-EEG as examples), we integrate both modalities into the model inversion algorithm to combine their information in the parameter estimation process and potentially improve clinical decision-making. Here we demonstrated two cases in which the EZN estimation from multimodal functional recordings were integrated. The first case estimated the EZN from simultaneous scalp-EEG and SEEG recordings under SEEG stimulation (Fig. 5a and Supplementary Fig. 4a). Here we mapped the same source signals to both the SEEG and scalp-EEG sensors. Two key added values arise from performing simultaneous SEEG and scalp-EEG recordings. First, the

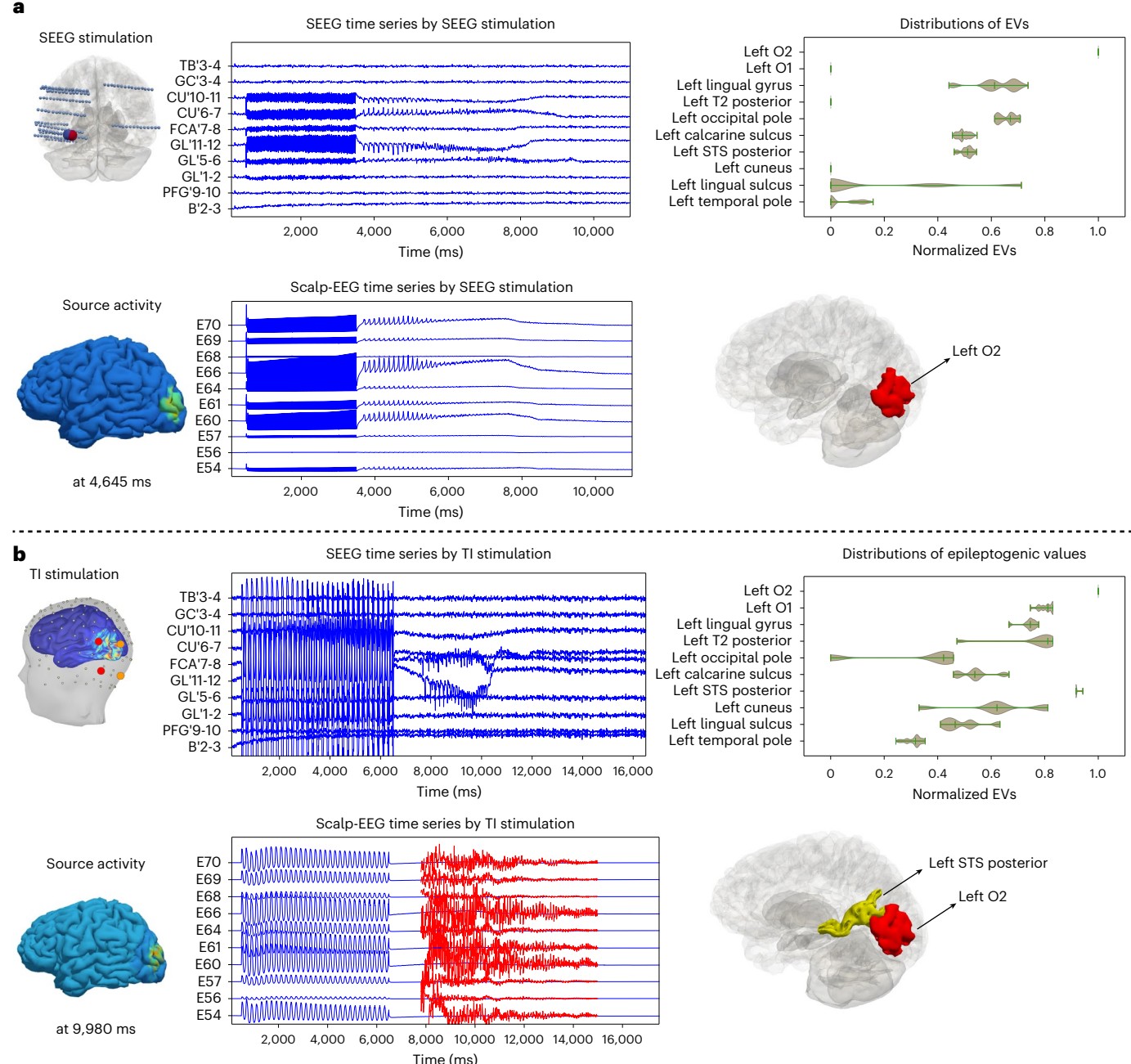

**Fig. 5 | Multimodal inference for EZN estimation from simultaneous SEEG and scalp-EEG recordings (synthetic data). a**, We combined ictal recordings from SEEG (middle top) and scalp-EEG (middle bottom) induced by SEEG stimulation (left), using multimodal inference to obtain the distribution of EVs (right top). The results were mapped in the 3D brain. In this case, we obtained the ground-truth left O2 in red. The spatial distribution of source activity at 4,634 ms is shown at the bottom left. **b**, We combined ictal recordings from SEEG (middle top) and scalp-EEG (middle bottom) induced by TI stimulation (left), using multimodal inference to obtain the distribution of EVs (right top). The results were mapped in three dimensions. In this case, we obtained the ground-truth left O2 in red and additional brain region as left STS posterior in yellow. The spatial distribution of source activity at 9,980 ms is shown at the bottom left.

scalp-EEG provides a whole-brain sampling although it mainly measures weak cortical surface signals in contrast to the SEEG, which samples only partial brain space with relatively strong signals from deep structures as well. Second, this pipeline provides a way to evaluate the roles of the scalp-EEG so that we could design a less invasive SEEG implantation in the future. The posterior of the EVs from a simultaneous model inversion showed the left O2 as the EZN (Fig. 5a). The second case estimated the EZN from simultaneous scalp-EEG and SEEG recordings under TI stimulation (Fig. 5b and Supplementary Fig. 4b). The posterior of the EVs from the multimodal model inversion showed the left O2 as the EZN and the left superior temporal sulcus (STS) posterior as a second candidate.

To complement the cases, we utilized the multimodal model inversion module on SEEG recordings from SEEG stimulation (Fig. 5a) and scalp-EEG recording from TI stimulation (Fig. 5b) in Supplementary Information. The posterior of the EVs from the multimodal model inversion showed the left O2 as the EZN (Supplementary Fig. 4c). Notice that, in this case, the SEEG and scalp-EEG data here were not simultaneously recorded. The SEEG and TI stimulation induced the seizures with different initial conditions. Then results would be valid only with an assumption that the HMC sampled all feasible initial conditions. For comparison, in the fourth case, we pooled the posterior distributions of the EVs from the SEEG (in Fig. 3d under SEEG stimulation) with the

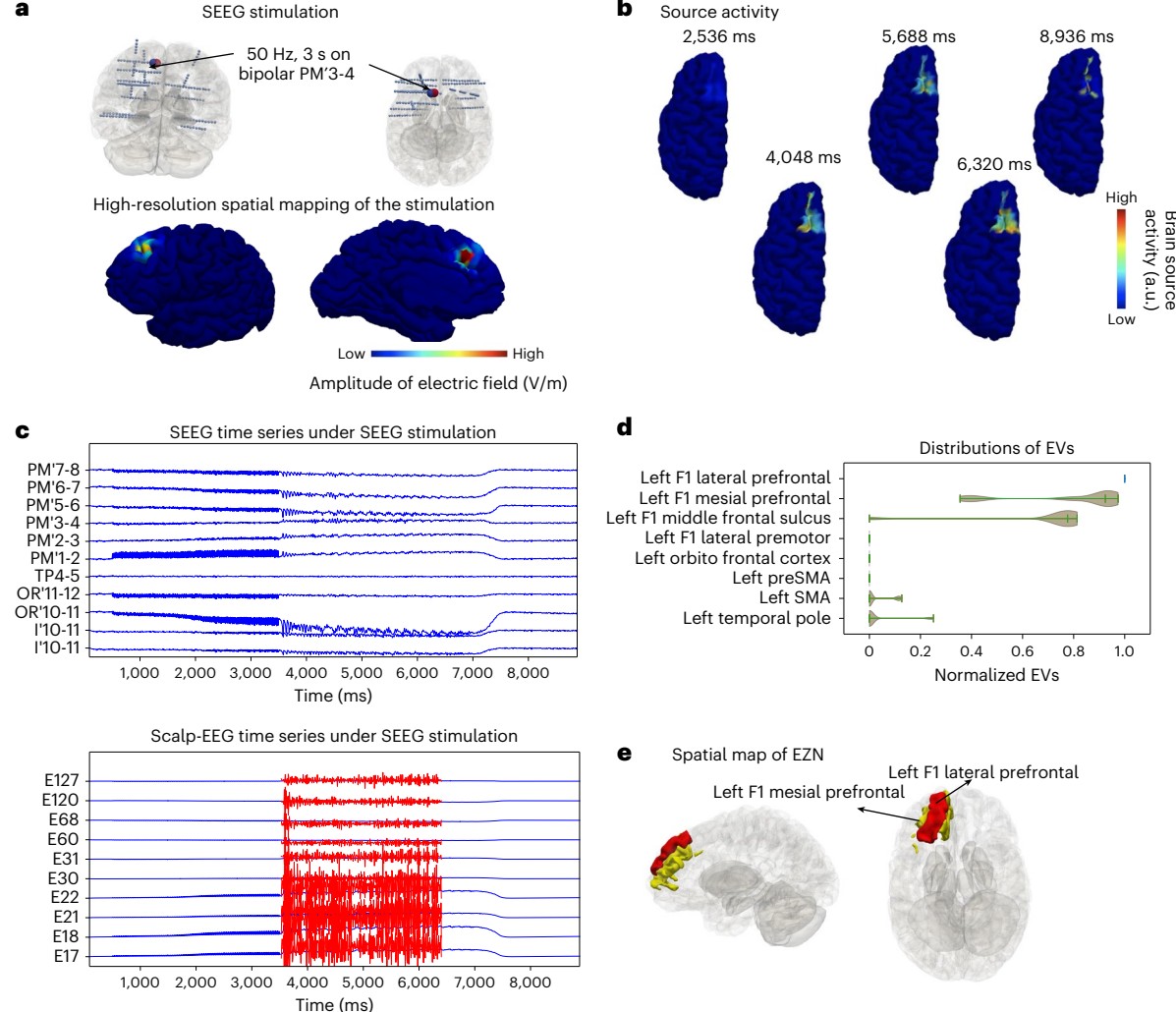

**Fig. 6 | Estimating EZN from SEEG stimulation for a second patient with frontal lobe epilepsy (synthetic data). a**, Top: PM'3-4 (large blue/red sphere) is the stimulated contact in the left frontal lobe, using a bipolar pulse stimulation (50 Hz for a duration of 3 s with pulse duration 1 ms). Bottom: spatial map showing the amplitude of electric field (shown in the color bar from low (left) to high (right) values) at each brain vertex, induced by SEEG stimulation (from sensor PM'3-4). **b**, Seizure dynamics were simulated using the Epileptor-Stimulation model and induced through SEEG stimulation. Neural source activity in arbitrary units (a.u.) is shown on the cortical mesh at five different time points, with values indicated by color. **c**, Selected simulated SEEG time series (top) and EEG time series (bottom) from SEEG-stimulation-induced seizure. The scaled-up time series during the seizure period are shown in red. **d**, Posterior distribution of EVs (higher value indicates higher chance for seizure) for eight selected regions obtained from the HMC sampling when analyzing simultaneously SEEG and scalp-EEG. SMA, supplementary motor area. **e**, The highest chance of being the EZN as left F1 lateral prefrontal in red mapped in the 3D brain and left F1 mesial prefrontal cortex and left-middle frontal sulcus in yellow.

EVs from the scalp-EEG (in Fig. 4d under TI stimulation). We obtained the left O2 as the intersection of the two distributions to be the only brain region in the EZN (Supplementary Fig. 4d).

**Validation of the pipeline**
To validate the pipeline, we added the results of a second patient with left frontal lobe epilepsy. The patient underwent resective surgery resulting in seizure freedom (surgery outcome class of Engel I[28]). Eight brain regions including the orbitofrontal, anterior cingulate, F1 lateral prefrontal, F2 rostral, superior frontal sulcus rostral, frontal pole and F1 mesial prefrontal cortices in the left hemisphere were resected. We built the high-resolution whole-brain model for two types of perturbation under the hypothesis that the left F1 lateral prefrontal cortex is the EZN. To model direct electrical stimulation by SEEG electrodes in the brain, we calculated the effect of the bipolar pulse perturbation (50 Hz, for a duration of 3 s on bipolar PM'3-4 electrode leads) (Fig. 6a). The seizure was induced around the left F1 lateral prefrontal cortex. From the

simulated neural source signals (Fig. 6b and Supplementary Video 3), we obtained simultaneous SEEG and scalp-EEG signals (Fig. 6c) using the source-to-sensor gain matrix. These synthetic SEEG and scalp-EEG signals during the ictal period served as input for multimodal inference to estimate the EZN. We obtained the left F1 lateral prefrontal cortex as the region with the highest EV from the HMC algorithm (Fig. 6d,e). The two neighboring regions (the left F1 measial prefrontal cortex and the left-middle frontal sulcus) were identified as secondary candidates that could belong to the EZN (Fig. 6d,e).

We also illustrated the results of this patient with TI stimulation in Extended Data Fig. 1. First, we used high-resolution modeling to simulate TI stimulation. We calculated the TI electric field, which is projected onto the cortical surface by stimulating two pairs of scalp-EEG electrodes (FPz–AF3 and AFz–Fp1 in Extended Data Fig.1a) at frequencies of 1,000 Hz and 1,005 Hz, respectively. The brain source signals on each vertex of the whole brain is shown in Extended Data Fig. 1b and Supplementary Video 4. We then mapped the neural activity from the

cortical surface onto the SEEG and scalp-EEG electrodes (Extended Data Fig. 1c) using the source-to-sensor mapping matrix. We extracted the data features from simultaneous synthetic ictal SEEG and scalp-EEG recordings (Extended Data Fig. 1c), which served as the input to the multimodal inference to estimate the EZN. We obtained the left-middle frontal cortex as the region with highest EVs from all samples (Extended Data Fig. 1d,e). The left F1 lateral prefrontal and left orbitofrontal sulcus were identified as the second candidates belonging to the EZN (Extended Data Fig. 1d,e).

We next demonstrated the role of heterogeneity of local connectivity. Here, local connectivity refers to how neuronal populations at different vertices are connected within a specific area or neighborhood. We introduced such heterogeneity in local connectivity under the hypothesis that within the EZN, local connectivity increases due to synaptic remodeling[31] or the loss of inhibitory synapses[32]. Extended Data Fig. 2 illustrates both source-level and sensor-level SEEG and EEG signals when local connectivity within the EZN is two to five times higher than that in other brain regions. Compared with the results obtained with homogeneous local connectivity (Fig. 6), only minor differences were observed during seizure periods. However, the neighborhood of the EZN showed a lower mean distribution of EVs from model inference, making it easier to identify the EZN.

Another application of the proposed pipeline is to test and mitigate false-positive stimulation results (non-habitual seizures being triggered) through a carefully designed protocol and optimized simulation and stimulation parameters. Here we systematically evaluated the effects of stimulation in terms of stimulation locations in two settings. (1) Within-electrode stimulation (PM'): stimulating all pairs of electrode leads within the same electrode using identical stimulation parameters (Supplementary Fig. 5). (2) Distributed cortical stimulation: stimulating ten additional SEEG bipolar pairs implanted in ten different cortical regions (Supplementary Fig. 6). Our simulations demonstrated that stimulation does not induce seizures in healthy regions. The theoretical basis for this lies in the seizure threshold of each region, which is directly related to its epileptogenicity. The EZN has a lower seizure threshold compared with healthy regions. This parameterization ensures that weak stimuli—based on clinically applied stimulation parameters—can trigger seizures in the EZN but not in healthy tissue.

The choice of atlas is an important topic in neuroscience studies, specifically for whole-brain network studies, and should be aligned with the specific research objectives. We selected the VEP atlas[27] because it is purposefully designed for epileptology, considering both anatomical and functional features within each structure, particularly in relation to EZN and surgical applications. Moreover, we extended our pipeline to be easily adaptable to other atlases, using the Desikan–Killiany (DK) atlas[33,34] as an example. We demonstrated this application using the DK atlas while maintaining the same simulation parameters as those used for the VEP atlas, including the stimulation electric field in Supplementary Fig. 7. The simulation of source activity and SEEG and scalp-EEG remains similar, particularly in terms of spatiotemporal patterns. This is because the primary difference arises from variations in the size of the EZN and structural connectivity (Supplementary Fig. 7a–c). The forward solution remains unchanged owing to the high-resolution stimulation at the source level. However, the estimated EZN is larger in the DK atlas because its brain regions are more extensive compared with those in the VEP atlas. As an example, we examined a case where the EZN network in the superior frontal gyrus is substantially larger in the DK atlas than in the VEP atlas. In the VEP atlas, the superior frontal gyrus is subdivided into five distinct structures.

A detailed quantitative comparison of SEEG and TI stimulation effects—characterized by spatial distribution and activation amplitude—is provided in Supplementary Information section 'Quantification of SEEG and TI stimulation effects' and Supplementary Fig. 8.

## Discussion

The first application of this pipeline can help us to answer a crucial diagnostic question: how to infer the underlying EZNs from stimulated seizures? To answer this question, it is necessary to investigate the following problems, illustrated in Supplementary Fig. 9. First, what is the proper range of stimulation parameters to induce seizures for diagnosis? Currently, no standardized stimulation protocols exist[20]. The French guidelines suggest parameters for both low-frequency (1 Hz) and high-frequency (50 Hz) stimulation with specified ranges on pulse width, pulse intensity and stimulation duration[3]. Our personalized whole-brain model has the ability to predict the stimulation effect for individual patients given specified stimulation parameters and should be validated in future empirical studies. On the patient-specific level, among the proper range of stimulation parameters, then what are the optimal stimulation parameters for inducing seizures for diagnosis? Second, what is the relationship between the EZN triggered by stimulation and the one occurring during spontaneous seizures? More specifically, we want to investigate whether the EZN estimated from stimulation-induced seizures equals the EZN of spontaneous seizures. It could be possible that stimulation forces seizures in brain regions that would not be epileptogenic otherwise, or that stimulation uncovers only a subset of epileptogenic regions. Even for spontaneous seizures, the limited observations of recordings of spontaneous seizures may also tell only part of the story. Third, how many seizures induced by SEEG stimulation are sufficient for identifying the full underlying EZN? We can investigate this question by systematically evaluating the EZNs on SEEG-stimulation-induced seizures generated by virtual brain twin models with given EZNs. The proposed VEP-stimulation pipeline with high-resolution modeling provides a reasonable framework to address all these questions.

The diagnostic pipeline for TI stimulation can provide a theoretical basis for non-invasive diagnosis and treatment of epilepsy. TI stimulation is a good candidate for both diagnosis and treatment in epilepsy within the non-invasive stimulation family. Other non-invasive stimulation techniques, such as tDCS[35], are mostly used for treatment only in epilepsy. The key feature of TI stimulation is that it enables a focal stimulation that can reach deep brain structures non-invasively[21,36], such as the hippocampus, which quite often is involved in the EZN in temporal lobe epilepsy. TI stimulation has evoked seizure-like events in the mouse hippocampus that have been experimentally well controlled[37] and strategies exist to replicate high-suprathreshold electric-field values in humans[38]. Our study presents a pipeline for predicting the use of TI in diagnosing the EZN in human epilepsy, and lays the foundation for optimizing TI-stimulation strategies. This framework can be further extended to build high-resolution, patient-specific models for future therapeutic interventions.

To achieve non-invasive diagnosis and treatment with TI in the future, we need to optimize the workflow using purely non-invasive data. A key next step, directly building on this work, is to systematically estimate the EZN from scalp-EEG signals within a cohort of patients. In the presented example, we obtained a good estimate of the EZN from non-invasive scalp-EEG signals, with localization in superficial cortical gray matter. Future studies should assess the robustness of the VEP across different epilepsy types and EZN locations. In addition, the required spatial resolution (number of scalp-EEG recording channels) for accurately identifying the underlying EZN needs further investigation. Another future direction is to develop a non-invasive VEP using high-resolution magnetoencephalography (MEG) data. We have obtained preliminary results with $^{23}$Na MRI[39] and now need to develop methods for VEP estimation and reconstruction from high-resolution EEG and MEG data. A further question is how precisely the EZN can be identified using interictal spikes or network features from EEG and MEG—potentially allowing seizure-inducing stimulation to be avoided.

Virtual brain twins have the capability to integrate multimodal neuroimaging data, including anatomical recordings, such as T1-weighted

MRI, diffusion-weighted MRI and CT; as well as functional ones, such as scalp-EEG, MEG, SEEG, subdural grid, positron emission tomography and functional MRI. In this work, we extend this integration by designing a multimodal inference version of the HMC algorithm for simultaneous functional recordings of SEEG and scalp-EEG[40], and can be further extended to other combinations, including MEG and scalp-EEG[41], and SEEG and MEG[42]. Across functional recordings, the main distinction lies in the forward solution. The usage of this multimodal inference is also valid in cases where different recordings start with the same brain states, or when the sampling algorithms can cover all possible initial conditions.

We extended this pipeline by incorporating the capability to model heterogeneous local connectivity in a brain region-specific manner. Three key elements must be taken into account: (1) the specific EZN associated with different types of epilepsy, (2) the type of neural models applied, and (3) personalization. In virtual brain twins, local connectivity should be derived from the anatomical information, rather than from functional connectivity and effective connectivity[43]. Anatomical connectivity varies across different EZNs depending on the type of epilepsy. For example, increased local connectivity within the EZN may result from enhanced excitatory synapse formation driven by axonal sprouting[44,45] or from the loss of inhibitory synapses[32]. Conversely, neuronal loss, axonal damage and synaptic pruning may reduce anatomical connectivity, although the exact mechanisms remain unclear[46]. The choice of the neural activity model also has an important role in representing the heterogeneity of local connectivity. Our results indicate that incorporating local connectivity heterogeneity does not substantially alter brain activity or seizure propagation when using the Epileptor model in this pipeline. The Epileptor, as a phenomenological model, and its excitability parameter $x_0$ already account for the coexistence of increased excitatory connectivity and decreased inhibitory connectivity, which reflects the hyperexcitability of epileptogenic tissue. However, region-specific local connectivity may play a more important role in biophysical models[47,48]. Currently, personalized measurements of local connectivity are not feasible in routine clinical practice. However, advancements in neuroimaging techniques may provide such information for research purposes and, in the future, for clinical applications[49].

The current pipeline has several limitations. Model-based approaches are usually computationally expensive and parameter sensitive, which may pose challenges for real time use in clinical routine, such as the optimization of stimulation parameters. The current pipeline does not consider the dynamics of the epileptic disorders over long timescales nor the long-term effects of the stimulation. Ongoing systematic evaluations and studies will also improve the current pipeline in the following ways. High-resolution modeling of the subcortical structure and a more realistic forward resolution may improve the whole-brain modeling. The introduction of regional variations (such as cell density and receptor density)[50,51] may provide a fundamental way to improve the current pipeline.

The virtual brain twin concept in stimulation here can be extended to other brain disorders: Parkinson's disease[52] and schizophrenia[17]. Studies have demonstrated the safety of TI in human participants[53,54], and explored its application in brain disorders, such as epilepsy[24,55], Alzheimer's disease[56], Parkinson's disease[57] and essential tremor[58]. For broader applications across different brain disorders, selecting an appropriate neural mass model and key parameters is a key challenge. Furthermore, as the choice of the right atlas matters, the pipeline illustrates the application to different atlases. The proposed pipeline can be adapted for alternative stimulation techniques, such as deep brain stimulation[59] and other non-invasive modalities such as tDCS[35] and tACS[23]. These stimulation methods are commonly used for therapeutic interventions, and a key challenge lies in refining and adapting the models to account for both excitatory and inhibitory effects. Furthermore, incorporating neuromodulation into the modeling process is crucial for accurately capturing these effects.

## Methods

### Patient data

We used the data from two patients who underwent a standard pre-surgical protocol at La Timone Hospital in Marseille. The first patient is a 23-year-old female diagnosed with left occipital lobe epilepsy. She underwent resective surgery and was nearly seizure-free after surgery, with an Engel Class II outcome. The second patient is a 19-year-old male with left frontal lobe epilepsy. He underwent resective surgery resulting in complete seizure freedom and an Engel Class I outcome. Informed written consent was obtained in compliance with the ethical requirements of the Declaration of Helsinki and the study protocol was approved by the local Ethics Committee (Comité de Protection des Personnes sud Méditerranée 1). Each patient underwent a comprehensive presurgical assessment, which included medical history, neuropsychological assessment, neurological examination, fluorodeoxyglucose positron emission tomography, high-resolution 3 T MRI, long-term scalp-EEG and invasive SEEG recordings. They received non-invasive T1-weighted imaging (MPRAGE sequence, repetition time 1.9 s, echo time 2.19 ms, voxel size $1.0 \times 1.0 \times 1.0$ mm$^3$) and diffusion-weighted MRI images (with an angular gradient set of 64 directions, repetition time 10.7 s, echo time 95 ms, voxel size $1.95 \times 1.95 \times 2.0$ mm$^3$, b-weighting of 1,000 s mm$^{-2}$.) The images were acquired on a Siemens Magnetom Verio 3 T MR-scanner. The patients had invasive SEEG recordings obtained by implanting multiple-depth electrodes, each containing 10–18 contacts 2 mm long and separated by 1.5 mm or 5 mm gaps. The SEEG signals were acquired on a 128 channel Deltamed/Natus system. After the electrode implantation, a cranial CT scan was performed to obtain the location of the implanted electrodes.

### Data processing

To construct the individual brain network models, we first preprocessed the T1-MRI and diffusion-weighted MRI data. Volumetric segmentation and cortical surface reconstruction were from the patient-specific T1-MRI data using the recon-all pipeline of the FreeSurfer software package http://surfer.nmr.mgh.harvard.edu. The cortical surface was parcellated according to the VEP atlas[27], the code for which is available at https://github.com/HuifangWang/VEP_atlas_shared.git. The reasons for choosing the VEP atlas are as follows. (1) The VEP atlas is specifically designed for epileptology, incorporating anatomical and functional features of each brain structure, particularly in relation to the EZN and surgical applications. (2) The geometric features and sizes of the brain regions are well suited for both clinical applications and modeling, including model inversion. (3) Brain regions can be automatically labeled and personalized from T1-MRI scans using geometric and neuroanatomical information. (4) The VEP atlas has been clinically evaluated in a retrospective study including 53 patients, and is currently being assessed in another prospective trial with 356 patients, ensuring its clinical suitability and reliability[14–16]. A brief summary of the steps to obtain the VEP atlas goes as follows. T1-weighted images are processed using FreeSurfer to remove non-brain tissue[60], segment subcortical gray matter structures[61], normalize intensity[62] and generate cortical surfaces[63]. These surfaces are inflated, registered to a spherical template and corrected for topology[61,64]. The cortex is then subdivided into regions based on gyral and sulcal structures[65], forming the basis for constructing the VEP atlas. This construction involves splitting, merging and renaming operations to both cortical and subcortical regions. Cortical regions are divided based on the triangulated surface mesh, while subcortical regions are split directly on voxels. Nonlinear splits are applied in specific areas with high curvature, such as the callosal sulcus, whereas the superior frontal gyrus is split using a combination of linear and specialized methods based on cortical surface geometry[27].

We used the MRtrix software package to process the diffusion-weighted MRI[66], employing the iterative algorithm described in ref. [67] to estimate the response functions and subsequently used constrained spherical deconvolution[68] to derive the fiber orientation

distribution functions. The iFOD2 algorithm[69] was used to sample 15 million tracts. The structural connectome was constructed by assigning and counting the streamlines to and from each VEP brain region. The diagonal entries of the connectome matrix were set to zero to exclude self-connections within areas and the matrix was normalized so that the maximum value was equal to one. We obtained the location of the SEEG contacts from post-implantation CT scans using GARDEL (Graphical user interface for Automatic Registration and Depth Electrodes Localization), which is part of the EpiTools software package[70]. Then we co-registered the contact positions from the CT scan space to the T1-MRI scan space for this patient.

### High-resolution simulation of the Epileptor-Stimulation model

We modeled epileptic seizures on a patient-specific high-resolution brain model. There is no consensus about the precise biophysical mechanism of ion exchanges that leads to seizure onset. Previous studies have demonstrated an increase in excitability, spike frequency and oscillation power[71–74] when an external perturbation was applied. Experimental studies have linked seizure onset to specific changes in ion dynamics such as extracellular potassium or intracellular chloride[75–77]. We hypothesized that exposure of repetitive stimulation/pertubation generates a slow accumulation effect, which can push the system to the seizure state when a seizure threshold is reached.

We model this accumulation effect phenomenologically via the parameter $m$, which influences the oscillatory dynamics in the seizure onset state in the Epileptor model[78,79]. We render $m$ time dependent, $m = m(t)$, and introduce an exponentially decaying memory kernel of length $r$ through a linear ordinary differential equation, which slowly builds up when an external stimulus is applied until the seizure threshold is reached. In the absence of a stimulus, $m(t)$ slowly returns back to baseline. $H$ is the Heaviside function. When $m$ is greater than the seizure threshold $m_{thresh}$, the $H$ function equals 1, which kicks the system into the seizure-like state. Otherwise, the $H$ function equals 0.

The pial surfaces of both hemispheres define the spatial domain along which the network activity can unfold. Neural dynamics are governed by an extension of the original Epileptor model[26]. The extended Epileptor-Stimulation model includes a stimulation-accumulating state variable $m$ that can destabilize the system and produce a seizure. The neural activity at every vertex $i = 1, …, N$ of the network is governed by the following equations:

$$\dot{x}_i = y_i - f_1(x_i) - z_i + I_{ext} + I_{stim}$$
$$+\gamma_{lc} \sum_{j=1}^{N} S(g_{i,j})H(x_j, \theta_{lc}) + \gamma_{gc} \sum_{l=1}^{L} W_{k,l}H(X_l, \theta_{gc})$$
$$\dot{y}_i = c - dx_i^2 - y_i$$
$$\dot{z}_i = r(4(x_i - x_0 + nH(m_i - m_{thresh})) - z_i) + f_2(z_i)$$
$$\dot{m}_i = r(I_{stim} - m_i)$$

(1)

where:

$$f_1(x_i) = \begin{cases} ax_i^3 - bx_i^2 & \text{if } x_i < 0 \\ -(m_i + 0.6(z_i - 4)^2)x_i & \text{if } x_i \geq 0 \end{cases}$$

$$f_2(z_i) = \begin{cases} -0.1z_i^7 & \text{if } z_i < 0 \\ 0 & \text{if } z_i \geq 0 \end{cases}$$

(2)

$$H(x, \theta) = \begin{cases} 1 & \text{if } x \geq \theta \\ 0, & \text{if } x < \theta \end{cases}$$

$$S(x) = \tfrac{1}{2}e^{-|x|}$$

The state variables $x$ and $y$ describe the activity of neural populations on a fast timescale and can model fast discharges. The oscillation

of the slow permittivity variable $z$ drives the system autonomously between ictal and interictal states. The parameter $x_0$ indicates the degree of excitability and directly controls the dynamics of the neural population to produce seizures or not. As $m(t)$ plays an important role in the model with stimulation, Supplementary Fig. 10 demonstrates how the system behavior changes as $m$ varies. As $m$ increases from 0 to 1.5, which is below the threshold ($m_{thresh} = 1.8$), the oscillation in the upstate changes from stable spiral to a spiral with limit cycles (Supplementary Fig. 10a–d). Higher values of $m$ correspond to larger ranges of limit cycles in both dimensions, $x$ and $z$. When $m = 2.0$, which exceeds $m_{thresh} = 1.8$, $m$ directly influences $z$, pushing the system into the seizing state. The variable $m(t)$ is directly related to the excitability of the corresponding tissues. Tissue excitability in epileptogenic networks arises from a combination of factors, including ion channel dysfunction, an imbalance between excitation and inhibition, and altered ion homeostasis. However, care must be taken not to overinterpret its role, as we are working with a phenomenological model. There are two slow variables, $z(t)$ and $m(t)$. While $z$ drives the gradual changes underlying seizure initiation and termination, $m$ increases by accumulating the stimulation effects through the influence of the electric field. As $m$ increases, the stability of the system changes. Once $m$ reaches the threshold value $m_{thresh}$, it directly pushes $z$ into a state that prepares the system to transition into a seizure state.

Each vertex $i$ is locally connected to its neighbors through local connectivity and globally connected to other parts of the brain through global connectivity. Local connections, scaled by $\gamma_{lc}$, are described by a translationally invariant Laplacian coupling kernel $S(g_{i,j})$ where $g_{i,j}$ denotes the geodesic distance along the cortical mesh between vertices $i$ and $j$. The cortical mesh incorporates personalized geometric features of the cortical surface derived from an individual's T1-MRI. Global connections, scaled by $\gamma_{gc}$, along white matter fibers, are represented by the connectome matrix, where $W_{k,l}$ denotes the connection strength between brain areas $k$ and $l$. Each vertex $i$ is assigned to a brain area $k = 1, …, L$ according to a cortical parcellation. For this study, we used the VEP parcellation. The average neural activity $X_l$ of all vertices belonging to area $l$ is coupled throughout the network and projected uniformly to all vertices of area $k$. The default parameters of the system are $I_{ext} = 3.1$, $c = 1$, $d = 5$, $r = 1$, $a = 1$, $n = 1$, $m_{thresh} = 1.8$, $\gamma_{gc} = 0.1$, $\gamma_{lc} = 0.8$, $\theta_{gc} = -1$, and $\theta_{lc} = -1$. The time-varying input $I_{stim}$ describes the perturbation signal in each time step depending on stimulation parameters. To accomplish this, we computed the generated electric field from the two types of perturbation (1) an invasive SEEG stimulation and (2) a non-invasive TI stimulation. These are described in two separate subsections below.

### Calculation of the electric field of SEEG stimulation

The SEEG stimulation is applied to a pair of neighboring sensors, in which one acts as a cathode and the other one as an anode. This generates a bipolar pulse perturbation in the area where the electrodes are located. The parameters used clinically are restricted to frequencies of either 1 Hz or 50 Hz, weak amplitudes ranging from 0.1 mA to 3 mA, and pulse widths of 0.5–3 ms. In this paper, we used a frequency of 50 Hz, an amplitude of 3 mA, a pulse width of 2 ms, and a stimulation duration of 3.5 s and 3 s. The stimulus signal, $\psi$, is a waveform represented here as a biphasic pulse train of electrical current. The electric-field strength at each vertex depends on the distance between vertices and the stimulated pair of bipolar electrode leads. Shorter distances result in higher field strengths. To simplify, we identified the maximal field strength within each brain region and uniformly applied it to all vertices within that region, as shown at the bottom of Figs. 3a and 6a. We defined the $I_{stim}$ at each vertex as $\psi \times$ electric-field strength at this vertex, with an example shown in Supplementary Fig. 2. Please note that the effects of the stimuli on brain signals are dependent on multiple factors, including the nonlinearity of local dynamics, local and global connections, and the stimulation current.

## Calculation of the electric field of the TI stimulation

Non-invasive stimulation with TI is a recent method that aims to mimic deep brain stimulation's capabilities to be both focal and subcortical without being invasive. This is done using the assumption that higher kilohertz stimulation frequencies have an ignorable effect on neuronal activity[21,36]. To obtain the distribution of the TI field in the gray matter, we started by computing the electric-field distribution from the simulation of tDCS via SimNIBS[80]. First, we reprocessed the individual T1-weighted MRI via the SimNIBS process 'mri2mesh', which resulted in the segmentation into five head tissues: white and gray matters, cerebrospinal fluid, skull and scalp. This process creates the tetrahedral meshes necessary for the finite element method for the simulation of the electric-field distribution. We computed the tDCS field for each pair of electrodes positioned based on the location of the co-registered 128 electrodes scalp-EEG cap, using the 10-5 system. Circular electrodes with a diameter of 10 mm and a thickness of 5 mm were directly applied on the head. Then, we projected these electric fields onto the resampled Freesurfer surface (20,484 vertices). We extracted the amplitude of the electric field at each vertex of the cortex surface. These fields derived from the coupled electrodes pairs were projected onto the normal of the cortical surface, then modulated by multiplying them with two sinusoids with frequencies of 1,000 Hz and 1,005 Hz, respectively, for 1 s. Two oscillatory electric fields, which were linearly summed, provided an interference field. To obtain a correct representation of the high-frequency activity, the sampling rate was 30 kHz. We extracted the 5 Hz peak envelope by using spline interpolation. Finally, we included this envelope in the virtual brain model as the source of external simulation for the cortex vertices (20,484).

For example, for patient 1, we targeted the highest TI stimulation effect on the left O2 in the VEP atlas[27]. We used the optimization process in SimNIBS to identify the best 4 electrodes configurations (10-5 scalp-EEG montage), which maximized the electric field in the left O2. This procedure allowed us to identify, in our case, PPO3–PPO5 and P5h–PO5h (ref. 29) as the best electrode couple for the left O2 stimulation. Thus obtained electric-field strengths ranged up to 0.48 V m⁻¹ at the cortical target sites, consistent with previously published empirical and simulation studies[24,54]. These field strengths are probably too weak to induce seizures in biological tissue. Nonetheless, future advancements in multipolar TI—using more than two electrode pairs—could enable the generation of stronger, more focal electric fields at target locations while maintaining low applied electric currents[38,81]. For our demonstration cases, we rescaled the amplitude of the stimulus signal to be able to induce seizures in the phenomenological Epileptor model and used it as an input to the $I_{stim}$ parameter. A simulated time series of one selected vertex from the EZN is shown in Supplementary Fig. 3.

## Forward solution for SEEG signals

The forward solution for the SEEG signals maps the neural activity from the sources to the sensors (SEEG contacts), represented by a source-to-sensor matrix (gain matrix). As sources for our model, we used the vertices of the pial surface for the cortical regions, and each subcortial region as a single node as in the neural mass model (NMM). Surfaces are represented as triangular meshes. For the NMM, we defined the mapping $g_{j,k}$ from the source brain region $j$ to the sensor $k$ as the sum of the inverse of the squared Euclidean distance $d_{i,k}$ from vertex $i$ to sensor $k$, weighted by the area $a_i$ of the vertex on the surface.

$$g_{j,k} = \sum_{i=0}^{N_j} a_i/d_{i,k}^2$$

Here vertex $i$ belongs to region $j$ which has $N_j$ vertices in total. The area $a_i$ of vertex $i$ was obtained by summing up one-third of the area of all the neighboring triangles. Vertices belonging to the same brain region were summed to obtain the gain for a single region of our brain

network model. The resulting gain matrix has dimensions $M \times N$, with $M$ being the number of regions and $N$ the number of sensors.

Matrix multiplication of the simulated source activity with the gain matrix yielded the simulated SEEG signals, that is, $SEEG_k(t) = \sum_j g_{j,k} x_j(t)$, where $x_j(t)$ is the time series of the source-level signals. This distance-based approach and the summation of all the vertices within each region neglect the orientation of the underlying current dipoles. Pyramidal neurons, which are oriented normal to the cortical surface, are assumed to be the physiological source of any electric signal recorded with SEEG, scalp-EEG or MEG[82]. The direction of the dipolar momentum associated with the NMM at typical spatial resolutions of 10–20 cm² in virtual brain networks of 100–200 nodes[27]. However, approaches that use high-resolution representations of the network, allowing for the computation of a surface orthogonal, have this ability. To solve the forward problem, we followed the analytical solution proposed in ref. 83 for electric fields in an unbounded homogeneous medium. This choice of forward model assumes no boundary effects of changes of conductivity at tissue boundaries. A previous study has shown that the error of an unbounded homogeneous conductivity model compared with a more accurate finite element method model with changes in conductivity is relatively small for electric fields generated by dipoles deep in the brain and electrodes close to the source[84]. Therefore, we can estimate the gain matrix elements $g_{i,k}$ approach by

$$g_{i,k} = a_i/(4\pi\sigma)Q \cdot (r_k - r_i)/(|r_k - r_i|^3)$$

where $r_k$ and $r_i$ are the position vectors of sensor $k$ and source vertex $i$, respectively. $|v|$ represents the L2 norm of a vector $v$. $Q$ is the dipole orientation vector and $\sigma$ is the electric conductivity. As we assume constant conductivity across the brain, it becomes merely a scaling factor, which we set to $\sigma = 1$.

## Forward solution for scalp-EEG signals

To compute the forward solution of scalp-EEG signals, we first reconstructed three individual surfaces (inner skull, outer skull and head) of the patient based on boundary element models using Brainstorm[85]. Then in Brainstorm, we co-registered the scalp-EEG electrodes positions (of the Hydrocel E1 128 channels electrode cap) onto the head surface, according to the fiducial points of the patient's T1-MRI. We applied a slight manual correction to better orient the scalp-EEG cap to the individual anatomy. Finally, we derived an scalp-EEG forward model using a 3-shell boundary element model (conductivity, 0.33 S m⁻¹, 0.165 S m⁻¹, 0.33 S m⁻¹; ratio, 1/20)[86] and the OpenMEEG method implemented in Brainstorm[87,88], to provide a realistic head model. Then we obtained a gain value for each dipole (20,484 vertices), with the constrained direction normal to the cortex surface, for each scalp-EEG electrode. Finally, the gain matrix derived from the head model was multiplied by the simulated time series of the brain sources to obtain the scalp-EEG activity, that is, $EEG_l(t) = \sum_j g_{j,l}^{EEG} x_j(t)$, where $g_{j,l}^{EEG}$ from the source signal on brain region $j$ to the $EEG$ signal on channel $l$.

## Calculation of the SEEG and scalp-EEG data features

We extracted the data features from the SEEG and scalp-EEG signal to be the input of the model inversion modules. The SEEG data were re-referenced using a bipolar montage, which was obtained using the difference between two neighboring contacts on one electrode. The two-dimensional Epileptor model, introduced below, is suitable for fitting the envelope of the seizure time series. Ideally, the envelope follows a slightly smoothed rectangular function from the onset until the offset of the seizure. To get a well-formed target that our model should fit, we extracted the bipolarized SEEG signal from 10 s before the seizure onset until 10 s after the seizure offset. We identified the outlier time points that were greater than two times the standard deviation of the extracted signal and replaced them with the mean of the extracted signal. The signal was then high-pass filtered with a cut-off at

10 Hz to remove slow signal drifts. The envelope was calculated using a sliding-window approach with a window length of 100 time points. The signal inside the window was squared, averaged and log-transformed. From the resulting envelope, we again identified and removed outliers, as described above. Finally, the envelope was smoothed using a lowpass filter with a cut-off of 0.05 Hz. The mean across the first few seconds of the envelope was used to calculate a baseline, which was then subtracted from the envelope. The same procedure was used for the EEG data, the only difference being that we used absolute values for the gain matrix, to get the accumulated effect on each EEG electrode throughout the entire seizure.

### The HMC model inversion

By taking advantage of timescale separation and using averaging methods, the Epileptor can be reduced to a two-dimensional system[89]:

$$\dot{x}_i = I_1 - x_i^3 - 2x_i^2 - z_i$$

$$\dot{z}_i = (1/\tau_0)\left(4(x_i - x_{i,0}) - z_i + K\sum_{j=1}^{N} C_{i,j}(x_j - x_i)\right) \quad (3)$$

where $\tau_0$ scales the length of the seizure. The external input is defined as $I_1 = 3.1$. We used the two-dimensional Epileptor for the model inversion, that is, the parameter estimate of the model, from the scalp-EEG and SEEG recordings.

For the individual SEEG model inversion, the forward solution is: $\text{SEEG}_k(t) = \sum_j g_{j,k} x_j(t)$. For the individual scalp-EEG model inversion, the forward solution is: $\text{EEG}_l(t) = \sum_j g_{j,l}^{\text{EEG}} x_j(t)$.

For the multimodal model inversion, we projected the same source signals to the scalp-EEG and SEEG at the same time while we calculated the likelihood according to the HMC algorithm, that is, $\text{SEEG}_k(t) = \sum_j g_{j,k} x_j(t)$; $\text{EEG}_l(t) = \sum_j g_{j,l}^{\text{EEG}} x_j(t)$.

For the model inversion, we applied the No-U-Turn Sampler (NUTS), an adaptive variant of the HMC algorithm to sample the posterior density of the model parameters. The performance of the HMC is highly sensitive to the step size and number of steps in the leapfrog integrator for updating the position and momentum variables in a Hamiltonian dynamic simulation[90]. We used NUTS, which is implemented in Stan and extends HMC with adaptive tuning of both the step size and the number of steps in a leapfrog integration to sample efficiently from the posterior distributions[90,91]. To overcome the inefficiency in the exploration of the posterior distribution of the model parameters, we used a reparameterization of the model parameters based on the map function from the model configuration space to the observed measurements[10]. Our reparameterization-based approach reduces the computation time by providing more effective sample sizes and removing divergences by exploring the posterior distributions of the linear combinations of regional parameters that represent the eigenvectors obtained from the singular value decomposition of the gain matrix. We denote the matrix of the eigenvectors of $G^T G$ as $V$ and the new parameters as $x_{i,0}^* = V^T x_{i,0}$ and $z_i(t_0)^* = V^T z_i(t_0)$. We ran the model inversion on both empirical and simulated seizures with 16 chains starting from 8 optimized initial conditions. The eight optimized initial conditions are the output of MAP estimation algorithm[15]. We ran the MAP estimation algorithm 50 times and selected the best 8 results in terms of the likelihood. We assessed model identifiability based on an analysis of posterior samples, which demonstrated that the sampler explores all the modes in the parameter space efficiently. The analysis includes trace plots (evolution of parameter estimates from draws over the iterations), pair plots (to identify collinearity between variables) and autocorrelation plots (to measure the degree of correlation between draws of samples). Sampling convergence of the algorithms was assessed by estimating the potential scale reduction factor and calculating the effective sample size based on the samples of the posterior distributions, providing estimates of the efficient run times of the algorithm.

### Calculation of the EVs

Using the HMC model inversion algorithm, we obtained the estimated source time series, and based on these, we calculated brain region-specific EVs. We checked the source time series (variable $x$ of the two-dimensional Epileptor in equation (3)) of region $i$ for values above a threshold of 0. The first occurrence of such a value is considered to be the onset of the seizure $t_i$ in that region. We define $t_0 = \min(t_i), i = 1, \ldots, 162$. Brain regions with no estimated seizure (no values above 0) are assigned an onset value $t_i = 200$. The $\text{EV}_i$ for brain region $i$ is calculated as $\text{EV}_i = -\log(((t_i - t_0) + 1)/20)$. Then we normalized the EV vector to $[0, 1]$ for each sample and plotted the distribution of EVs.

### Reporting summary

Further information on research design is available in the Nature Portfolio Reporting Summary linked to this article.

### Data availability

The data used in this work are available via Code Ocean: the compute capsules can be accessed via https://doi.org/10.24433/CO.3316132.v1 (ref. 92; part 1) and https://doi.org/10.24433/CO.2143670.v1 (ref. 93; part 2). These data include a patient's preprocessed anatomical information and high-resolution functional simulation data, which can be used for relevant calculations and simulations presented in this paper. The patient raw datasets cannot be made publicly available due to the data protection concerns regarding potentially identifying and sensitive patient information. Interested researchers may access the datasets by contacting the corresponding authors and F.B. (Fabrice. BARTOLOMEI@ap-hm.fr). Source data for Figs. 2–6 and Extended Data Figs. 1 and 2 are available with this paper.

### Code availability

The code used in this work is available via Code Ocean: part 1, https://doi.org/10.24433/CO.3316132.v1 (ref. 92); part 2, https://doi.org/10.24433/CO.2143670.v1 (ref. 93); The codes are also available via GitHub at https://github.com/HuifangWang/VBT_INS_Stimulation.git.

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

## Acknowledgements

We thank M. Guye for providing the patient's anatomical data and F. Missey for the helpful discussions on TI stimulation. This work was funded through the EU's Horizon Europe Programme under grant number 101147319 (EBRAINS 2.0; V.J.) and number 101137289 (Virtual Brain Twin; V.J.), the Amidex Recherche Blanc under grant number AMX-22-RE-AB-135 (HR-VEP; H.E.W.), and agence Nationale de la Recherche under France 2030 bearing the reference ANR-24-RRII-0005 (NAUTILUS; V.J.), on funds administered by Inserm.

## Author contributions

V.J., H.E.W. and F.B. conceived of the study. H.E.W. and V.J. formulated and developed the methodology and provided funding acquisition. H.E.W., B.D., G.M.D., P.T. and J.-D.L. developed the algorithms and workflow, wrote the original draft and created visualizations. H.E.W., B.D., P.T., G.M.D., A.W., J.M. and J.-D.L. performed the data analysis. All authors contributed to the editing and revision of the paper.

## Competing interests

V.J., B.D., P.T., H.E.W. and A.W. hold a patent related to the technology and methods discussed in this article: A method and system for estimating an epileptogenic zone network: European Patent EP23169009.0.

## Additional information

**Extended data** is available for this paper at https://doi.org/10.1038/s43588-025-00841-6.

**Correspondence and requests for materials** should be addressed to Huifang E. Wang or Viktor Jirsa.

 

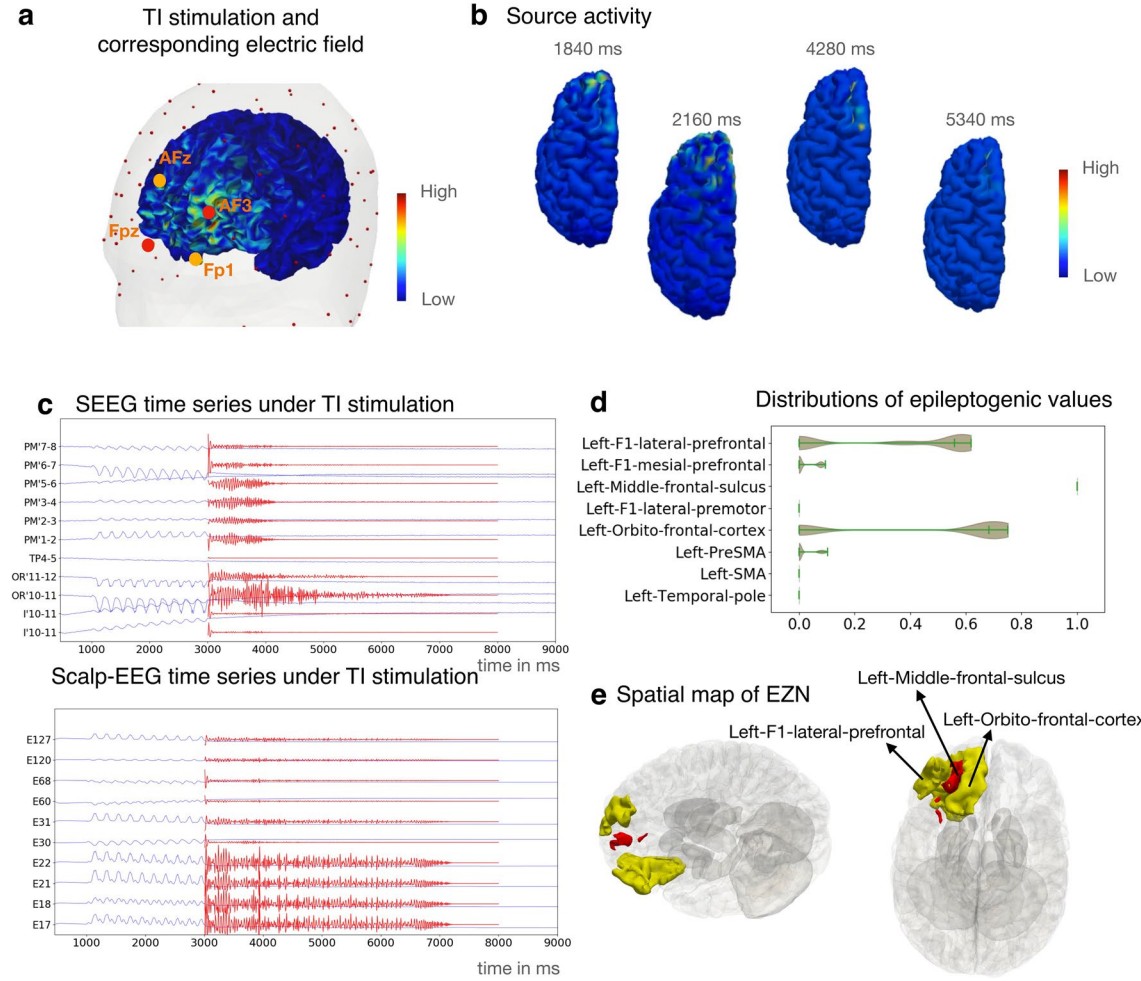

**Extended Data Fig. 1 | Estimating EZN from TI stimulation for a second patient with frontal lobe epilepsy (synthetic data). a** The electric field of TI stimulation by two pairs of scalp-EEG electrodes (shown in red and orange). We applied a frequency of 1000 and 1005 Hz for the first and second electrode pairs, respectively. The Spatial distribution of the peak activity of the TI envelope is colored on the 3D brain. **b** Seizure dynamics were simulated using the Epileptor-Stimulation model through the TI stimulation. Neural activity is shown on the cortical mesh at 4 different time points. **c** Selected synthetic SEEG time-series (top) and scalp-EEG time-series (bottom) from TI stimulation-induced seizure. The scaled-up time series during the seizure period are shown in red. **d** Posterior distribution of the EV (higher value indicates higher chance for seizure) for 8 selected regions the same as Fig. 6d. **e** The highest chance of being the EZN as left-middle-frontal-sulcus in red mapped in 3D brain and left-F1-lateral-prefrontal and left-orbito-frontal-cortex in yellow.

## a Source activity

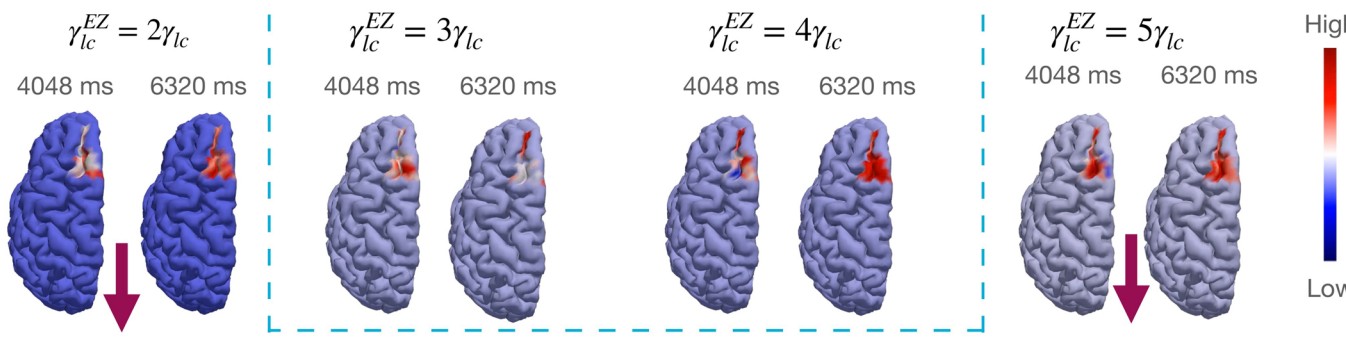

## b SEEG time series under SEEG stimulation

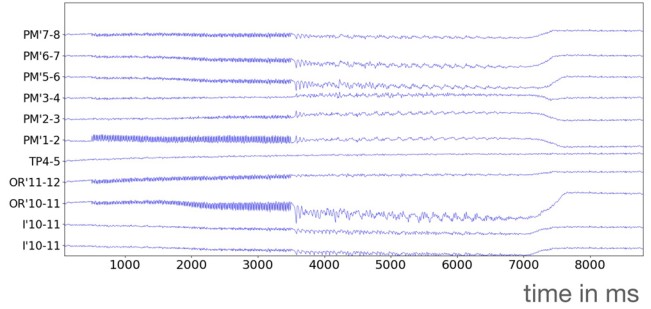

## Scalp-EEG time series under SEEG stimulation

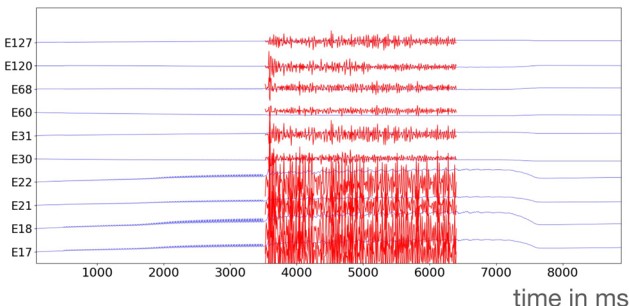

## Distributions of epileptogenic values

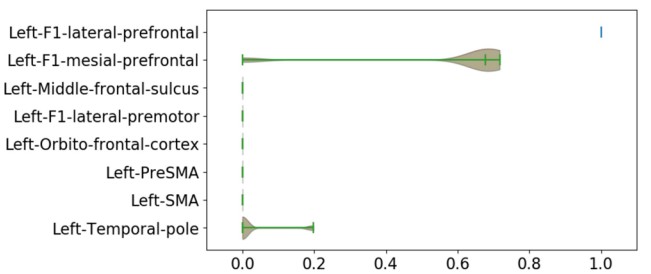

## c SEEG time series under SEEG stimulation

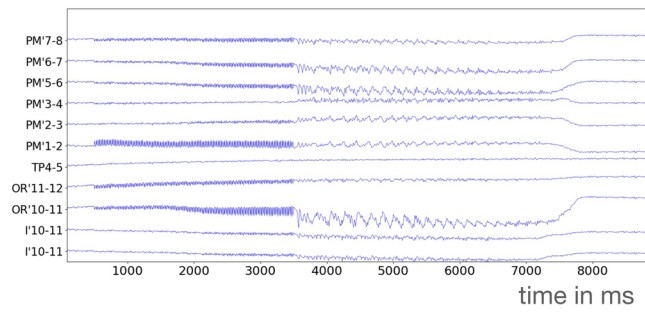

## Scalp-EEG time series under SEEG stimulation

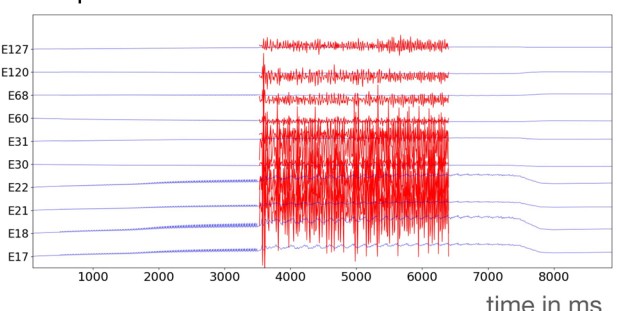

## Distributions of epileptogenic values

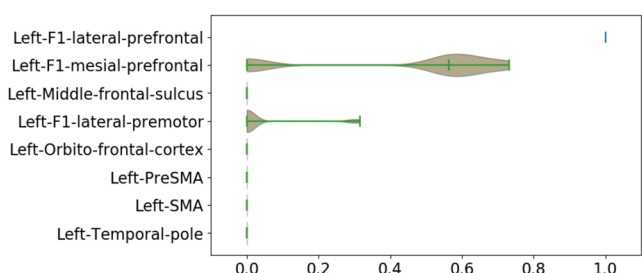

**Extended Data Fig. 2 | The effect of increasing local connectivity within EZNs under SEEG stimulation as shown in Fig. 6. a** Snapshot of source activity (corresponding videos available in the Supplementary Information) at two specified time instants for comparison when the local connectivity within EZN $\gamma_{lc}^{EZ}$ is $r$ times the local connectivity $\gamma_{lc}$ of other brain regions with $r = 2, 3, 4, 5$.

**b**. When $\gamma_{lc}^{EZ} = 2\gamma_{lc}$, selected simulated SEEG time-series (top) and EEG time-series (middle) from SEEG stimulation-induced seizure. The scaled-up time series during the seizure period are shown in red. Posterior distribution of EVs (bottom) from the HMC sampling are shown when analyzing simultaneously SEEG and scalp-EEG. **c** Same as **b**, but under the condition $\gamma_{lc}^{EZ} = 5\gamma_{lc}$.

# Reporting Summary

## Statistics

For all statistical analyses, confirm that the following items are present in the figure legend, table legend, main text, or Methods section.

| n/a | Confirmed | |
|---|---|---|
| ☐ | ☒ | The exact sample size (*n*) for each experimental group/condition, given as a discrete number and unit of measurement |
| ☐ | ☒ | A statement on whether measurements were taken from distinct samples or whether the same sample was measured repeatedly |
| ☒ | ☐ | The statistical test(s) used AND whether they are one- or two-sided *Only common tests should be described solely by name; describe more complex techniques in the Methods section.* |
| ☒ | ☐ | A description of all covariates tested |
| ☐ | ☒ | A description of any assumptions or corrections, such as tests of normality and adjustment for multiple comparisons |
| ☐ | ☒ | A full description of the statistical parameters including central tendency (e.g. means) or other basic estimates (e.g. regression coefficient) AND variation (e.g. standard deviation) or associated estimates of uncertainty (e.g. confidence intervals) |
| ☒ | ☐ | For null hypothesis testing, the test statistic (e.g. *F*, *t*, *r*) with confidence intervals, effect sizes, degrees of freedom and *P* value noted *Give P values as exact values whenever suitable.* |
| ☐ | ☒ | For Bayesian analysis, information on the choice of priors and Markov chain Monte Carlo settings |
| ☐ | ☒ | For hierarchical and complex designs, identification of the appropriate level for tests and full reporting of outcomes |
| ☒ | ☐ | Estimates of effect sizes (e.g. Cohen's *d*, Pearson's *r*), indicating how they were calculated |

*Our web collection on statistics for biologists contains articles on many of the points above.*

## Software and code

Policy information about availability of computer code

| Data collection | Siemens Magnetom Verio 3T MR-scanner for T1-weighted imaging and diffusion MRI images, Deltamed/Natus system for SEEG recordings. |
|---|---|
| Data analysis | FreeSurfer 6. 3.0: Used for volumetric segmentation and cortical surface reconstruction; https://surfer.nmr.mgh.harvard.edu/fswiki/DownloadAndInstall<br>Cortical surface parcellation: https://github.com/HuifangWang/VEP_atlas_shared.git<br>Mrtrix 0.3.16 software package for processing DW-MRI data. https://mrtrix.readthedocs.io/en/0.3.16/<br>GARDEL 1.0 (Graphical user interface for Automatic Registration and Depth Electrodes Localization) for location of the SEEG contacts from post-implantation CT scans. https://meg.univ-amu.fr/doku.php?id=epitools:gardel<br>SIMNIBS 4.0: Used for electric field calculation.; https://simnibs.github.io/simnibs/build/html/index.html<br>Brainstorm3:Forward solution for scalp-EEG signals; https://neuroimage.usc.edu/brainstorm/Installation#Requirements |

For manuscripts utilizing custom algorithms or software that are central to the research but not yet described in published literature, software must be made available to editors and reviewers. We strongly encourage code deposition in a community repository (e.g. GitHub). See the Nature Portfolio guidelines for submitting code & software for further information.

## Data

Policy information about availability of data

All manuscripts must include a data availability statement. This statement should provide the following information, where applicable:
- Accession codes, unique identifiers, or web links for publicly available datasets
- A description of any restrictions on data availability
- For clinical datasets or third party data, please ensure that the statement adheres to our policy

This work involves two types of data: raw clinical data, including T1-MRI and DWI-MRI images, CT scans, and SEEG data. The raw data are available upon request, but they are not central to this paper. All derived data and key information for personalized modeling will be publicly available after publication.

## Human research participants

Policy information about studies involving human research participants and Sex and Gender in Research.

| | |
|---|---|
| Reporting on sex and gender | We studied two patients:<br>1) A 23-year-old female diagnosed with left occipital lobe epilepsy. She underwent resective surgery and was nearly seizure-free post-surgery, with an Engel Class II outcome.<br>2) A 19-year-old male diagnosed with left frontal lobe epilepsy. He underwent resective surgery resulting in complete seizure freedom, with an Engel Class I outcome. |
| Population characteristics | We have two patients: a 23-year-old female with occipital lobe epilepsy and a 19-year-old male with frontal lobe epilepsy. One had a surgical outcome classified as Engel Class I, and the other as Engel Class II. |
| Recruitment | We used two epilepsy patients with drug resistant focal epilepsy who underwent a standard presurgical protocol at La Timone hospital in Marseille. We selected these two patients to validate our workflow. Future scientific studies should include a broader range of epilepsy types and surgical outcomes. |
| Ethics oversight | Informed written consent was obtained for all patients in compliance with the ethical requirements of the Declaration of Helsinki and the study protocol was approved by the local Ethics Committee<br>(Comité de Protection des Personnes sud Méditerranée 1) |

Note that full information on the approval of the study protocol must also be provided in the manuscript.

# Field-specific reporting

Please select the one below that is the best fit for your research. If you are not sure, read the appropriate sections before making your selection.

☒ Life sciences  ☐ Behavioural & social sciences  ☐ Ecological, evolutionary & environmental sciences

For a reference copy of the document with all sections, see nature.com/documents/nr-reporting-summary-flat.pdf

# Life sciences study design

All studies must disclose on these points even when the disclosure is negative.

| | |
|---|---|
| Sample size | This is a methodology and concept paper. We selected two patients with drug-resistant focal epilepsy. These two patients have different diagnoses and surgical outcomes. For each patient, the dataset includes both anatomical and functional data from multiple recordings, such as T1-weighted MRI, CT, diffusion-weighted MRI, and multiple stereo-EEG sessions. Since this is a methodology and proof-of-concept paper for personalized medicine, we believe that an in-depth analysis of two patients is sufficient to demonstrate the feasibility of our approach. |
| Data exclusions | No data were excluded from the analyses |
| Replication | The results are replicable if the same parameters were used on the same datasets.The results are replicable when the same parameters are applied to the same datasets. Since this study focuses on personalized virtual brain twins, replication is expected during the construction of virtual twins or the identification of the epileptogenic zone networks in similar cases. |
| Randomization | Because this study focuses on personalized medicine using patient-specific data, randomization is not necessary. |
| Blinding | The VEP analysis is independent of the patients' clinical hypothesis and surgery outcomes. Blinding was not applicable in this context for several reasons. First, the study is retrospective and all clinical interventions had already been completed prior to analysis. Second, the Virtual Epileptic Patient (VEP) analysis is computational and data-driven, conducted independently of clinical outcomes or hypotheses. Finally, since the methodology relies on objective modeling of individualized brain dynamics rather than subjective interpretation, the lack of blinding does not compromise the validity or integrity of the results. |

# Reporting for specific materials, systems and methods

We require information from authors about some types of materials, experimental systems and methods used in many studies. Here, indicate whether each material, system or method listed is relevant to your study. If you are not sure if a list item applies to your research, read the appropriate section before selecting a response.

## Materials & experimental systems

| n/a | Involved in the study |
|-----|----------------------|
| ☒ | Antibodies |
| ☒ | Eukaryotic cell lines |
| ☒ | Palaeontology and archaeology |
| ☒ | Animals and other organisms |
| ☐ ☒ | Clinical data |
| ☒ | Dual use research of concern |

## Methods

| n/a | Involved in the study |
|-----|----------------------|
| ☒ | ChIP-seq |
| ☒ | Flow cytometry |
| ☐ ☒ | MRI-based neuroimaging |

## Clinical data

Policy information about clinical studies

All manuscripts should comply with the ICMJE guidelines for publication of clinical research and a completed CONSORT checklist must be included with all submissions.

| | |
|---|---|
| Clinical trial registration | It is not a clinical trial. The clinical data is being used for a research study. |
| Study protocol | The goal of the study is to develop and evaluate the VEP high-resolution workflow for stimulation. VEP is a personalized workflow, so we selected two patients with drug-resistant focal epilepsy, each with different diagnoses and surgical outcomes. Using these patients' data, we built personalized whole-brain models. These models can predict the effects of stimulation and can be further used to better estimate epileptic networks. |
| Data collection | For each patient, the dataset includes both anatomical and functional data from multiple recordings, such as T1-weighted MRI, CT, diffusion-weighted MRI, and multiple stereo-EEG sessions. |
| Outcomes | The VEP models can predict the effects of stimulation and can be further used to better estimate epileptic networks. |

## Magnetic resonance imaging

### Experimental design

| | |
|---|---|
| Design type | No functional MRI data was recorded, only structural and diffusion weighted images. |
| Design specifications | Not used. |
| Behavioral performance measures | Not used. |

### Acquisition

| | |
|---|---|
| Imaging type(s) | presurgical T1 weighted MRI, presurgical diffusion weighted MRI, post SEEG implantation CT scan |
| Field strength | 3 Tesla |
| Sequence & imaging parameters | MPRAGE sequence, repetition time = 1.9 or 2.3 s, echo time = 2.19 or 2.98 ms, voxel size 1.0 mm3, FoV full head CT scans FoV full head, voxel size around 0.4mm * 0.4mm * 0.6mm |
| Area of acquisition | Whole brain scan |

Diffusion MRI    ☒ Used    ☐ Not used

| | |
|---|---|
| Parameters | Either single shell, b-values = [0,1000], 64 directions or multi-shell, b-values = [0, 1400, 1800], 200 directions, no cardiac gating used |

### Preprocessing

| | |
|---|---|
| Preprocessing software | Freesurfer v6, FSL v6, MRtrix 0.3.16 |
| Normalization | No spatial normalization was used in this study as all processing, modeling and inference is done in the imaging space of each individual patient.<br>Only a linear registration was performed to align between patient specific T1, diffusion and CT images. |

March 2021

| Normalization template | Not used. |
|---|---|
| Noise and artifact removal | T1 weighted was processed using the recon-all pipeline from Freesurfer.<br>Diffusion weighted MRI was processed using the functionality of the MRtrix software package. |
| Volume censoring | No volume censoring performed. |

## Statistical modeling & inference

| Model type and settings | Not used. |
|---|---|
| Effect(s) tested | Not used. |

Specify type of analysis: ☒ Whole brain ☐ ROI-based ☐ Both

| Statistic type for inference<br>(See Eklund et al. 2016) | Not used. |
|---|---|
| Correction | Not used. |

## Models & analysis

| n/a | Involved in the study |
|---|---|
| ☒ ☐ | Functional and/or effective connectivity |
| ☒ ☐ | Graph analysis |
| ☒ ☐ | Multivariate modeling or predictive analysis |

