## [Peer Review file · Nature Computational Science]

Virtual brain twins for stimulation in epilepsy

Corresponding Author: Dr Huifang Wang

Version 0:

Decision Letter:

**** Please ensure you delete the link to your author homepage in this e-mail if you wish to forward it to your co-authors. ****

Dear Dr Wang,

Your manuscript "Virtual brain twins for stimulation in epilepsy" has now been seen by 3 referees, whose comments are appended below. You will see that while they find your work of interest, they have raised points that need to be addressed before we can make a decision on publication.

The referees' reports seem to be quite clear. Naturally, we will need you to address **all** of the points raised.

While we ask you to address all of the points raised, the following points need to be substantially worked on:

- To rule out the possibility that the model is producing type 1 errors by identifying most stimulated regions as epileptogenic, please deliver the same stimulus to an unrelated part of the cortex, and observing if the model still identifies it as an EZ.
- Please discuss limitations in using atlas other than VEP for the individual connectome construction.
- Some more details should be provided in this manuscript to aid the readers in understanding the implemented pipeline.
- Please discuss if the model is generalizable.
- It is mentioned that you map the stimulus signal onto the parcellated brain areas based on distance. What stimulus is being described needs to be mentioned.

Please use the following link to submit your revised manuscript and a point-by-point response to the referees' comments (which should be in a separate document to any cover letter):

Link Redacted

**** This url links to your confidential homepage and associated information about manuscripts you may have submitted or be reviewing for us. If you wish to forward this e-mail to co-authors, please delete this link to your homepage first. ****

To aid in the review process, we would appreciate it if you could also provide a copy of your manuscript files that indicates your revisions by making use of Track Changes or similar mark-up tools. Please also ensure that all correspondence is marked with your Nature Computational Science reference number in the subject line.

In addition, please make sure to upload a Word Document or LaTeX version of your text, to assist us in the editorial stage.

If you have any issues when updating your Code Ocean capsule during the revision process, please email the Code Ocean support team Cc'ing me.

To improve transparency in authorship, we request that all authors identified as 'corresponding author' on published papers create and link their Open Researcher and Contributor Identifier (ORCID) with their account on the Manuscript Tracking System (MTS), prior to acceptance. ORCID helps the scientific community achieve unambiguous attribution of all scholarly contributions. You can create and link your ORCID from the home page of the MTS by clicking on 'Modify my Springer Nature account'. For more information please visit www.springernature.com/orcid.

We hope to receive your revised paper within three weeks. If you cannot send it within this time, please let us know.

Best regards,

Ananya Rastogi, PhD
Senior Editor
Nature Computational Science

Reviewers comments:

Reviewer #1 (Remarks to the Author):

General:

Interventions for drug resistant focal epilepsy rely on the identification of the Epileptogenic Zone (EZ). The current approach involves using stereo EEG (sEEG) electrodes to record seizure activity. A complementary method is to induce seizures via electrical stimulation, either through sEEG electrodes, or via Temporal Interference (TI).

The authors have built on advances in whole brain modeling, individualized parameter estimation and phenomenological models of epilepsy to construct a virtual analog to this pipeline. Patient MRI and sEEG data are used to determine the surface geometry, long range connectivity, epileptogenicity and global network scaling. These are used as inputs to a neural mass model, with node dynamics being determined by the epileptor model, adapted for stimulation. Stimulus is given to the model via an sEEG to source mapping, and the resulting source activity is mapped onto a participant specific cortical surface. Bayesian inference is then applied to determine the EZ.

The authors present two case studies on whom the novel pipeline has been applied. They report that the EZ determined by the model corresponds to the areas whose removal via resective surgery lead to a reduction in epileptic seizures.

Overall, the work is of very high importance and impact. Nonetheless, I also find some serious issues that need addressal.

Major Issues

Methodological

1. Activity is simulated at each vertex (20284), however heterogenous coupling is only used for connections between regions in the VEP parcellation (162). The local coupling is distance dependent and is scaled by a factor γ_{hom} which is invariant across ROIs. Considering that increased connectivity within the EZ is observed in focal epilepsy, homogenous local coupling may be a strong assumption.
2. The site of stimulation used in the virtual pipeline was determined based on the regions resected during surgery, and appears to be proximal to the previously identified EZ. It is possible that the model is producing type 1 errors by identifying most stimulated regions as epileptogenic. To rule out this possibility, I suggest delivering the same stimulus to an unrelated part of the cortex, and observing if the model still identifies it as an EZ.

Future Directions

3. Can the epileptor model parameters be accurately estimated from surface EEG alone? If so, this method coupled with Temporal Interference could provide an entirely non-invasive treatment for drug resistant epilepsy.
4. Allowing γ_{hom} to vary across ROIs and determining its values for each participant could add another dimension to the subject specificity of the treatment and improve the method's accuracy.

Minors

Formatting

1. Figure 1: The plots within sub figures B, E, F have axis labels and ticks that are illegible at this scale and create visual clutter.
2. Line 114; Local and global propagation of seizures across vertices is mentioned separately, ostensibly because the local propagation occurs under homogeneous coupling. However, since that distinction is only made later in the article the phrasing here seems awkward.
3. Line 161; Consider replacing "with the hypothesis" with "to test the hypothesis".
4. Figure 7: In the caption, the heading for subfigure E is not bold, as the other headings are.

Reviewer #1 (Remarks on code availability):

Codes are fine. readme files are in place for guidance that are adequately commented

Reviewer #2 (Remarks to the Author):

The present study attempts to use the virtual brain twin concept based on their previous work using VEP models and frameworks. The Virtual brain twins are personalized, generative, and adaptive brain models based on data from an individual patient's brain for scientific and clinical use. In this study, they introduce the first high-resolution virtual brain twin workflow, specifically designed for estimating the EZN involving direct electrical stimulation through SEEG electrodes. Overall, I think this work is important and has the potential to advance individual-specific brain mapping and identification of EZN estimation under a variety of invasive and non-invasive clinical settings. However, I have a few fundamental queries related to workflow implementations and the translation from invasive to non-invasive settings. Some more major and minor comments are outlined below. I sincerely hope that addressing these may improve the clarity of the methods and presentation of the findings in the revised version of this manuscript.

In the data processing step to construct individual brain network models, the authors used parcellation of the cortex using a VEP atlas. Are there any limitations in using other atlas for the individual connectome construction?

It is perhaps not clear what the scientific rationales are for choosing the VEP atlas for the individual connectome of the patients over other existing atlases.

VEP atlas is derived from their previous study the code/script and reference are included. However, I feel some more details should be provided in this manuscript to aid the readers in understanding the implemented pipeline.

Are the results of the stimulation sensitive to the choice of the VEP atlas and location of the brain regions determined by the atlas, or is it generalizable and the stimulation results based on SEEG/Scalp EEG data are reproducible regardless of the choice of this atlas to construct virtual brain twins? Perhaps the authors can explain this in more detail.

The authors mentioned that exposure to repetitive stimulation/perturbation generates a slow accumulation effect, which can push the system to the seizure state when a seizure threshold is reached. Beyond this excitability threshold, a network of neurons likely exhibits a high firing rate and increase in spiking activity.

The authors use this logic to introduce a slow variable that mimics the accumulation effect in the mathematical model. Mathematically, the time-dependent accumulation variable $m(t)$ in the model is well described and acts like a switch regarding perturbations introduced by brain stimulations. However, $m(t)$ and its relationship with a fast-spiking variable need to be defined better, and the link of $m(t)$ to underlying biophysics should be better described. Additionally, we know that perturbation effects are very specific and depend on various factors, the magnitude of the current pulse, pulse width, electric field distributions, and stimulation frequency tailored to the subject's response to field oscillations, Excitation-Inhibition balance (Gaba-Glutamate ratio). Those factors have substantially differential effects on the excitability of a network of spiking neurons. Hence, in their phenomenological model, what exactly accumulates as represented by the slow variable here is a bit fuzzy. How these accumulation time scales relate to the dynamics of fast variables could be important for understanding the entry to the seizure state once an appropriate threshold is reached. I find this in the current version of the implementation, and the description is unclear.

The author mentions local connections are heterogeneous; however, to include that in the virtual brain twin model is nearly impossible at this stage

as the pieces of information needed on interneurons and dendritic arborization in clinical measurements have yet to be made available. However, they only consider geometric features of cortical surfaces for the epileptic simulations. It is again unclear what geometric features of cortical surface they are talking about. Do they mean dipole orientation and magnitude scaling as a function of the underlying source location that goes into LED field matrix calculations for SEEG/EEG stimulations? As I understand, they are only using an individual connectome, as shown in the workflow diagram in Figure 1, which incorporates streamlined counts and fibre orientation density measured from DW-MRI.

The author says we map the stimulus signal onto the parcellated brain areas based on distance. What stimulus they are describing needs to be mentioned. Perhaps that information is important for the readers.

This is related to the earlier question as well. If I understand their pipeline, the authors have created a virtual brain twin based on the connectivity estimate from patients' diffusion imaging data. Once they have the virtual brain twin of the individual patient, they will apply a relatively new tACS stimulation technique where vectorial tACS fields are added together, resulting in a T1 field estimate for the brain regions. Subsequently, they used this estimated field as an input to the I_{stim} parameter and simulated their phenomenological model on one node/vertex on the connectome (from the identified EZ). However, my question is whether the stimulation protocol or, in other words, the T1 field needs to be tailored in an individual/patient-specific manner or whether the individual specificity only lies in the virtual brain twin constructed in the pipeline based on the individual connectome. My conceptual understanding and assumption are that individual connectome requires individual stimulation protocol likely for the most effective clinical outcome of the adjuvant treatment strategy. Please explain what I am missing here.

For the model inversion using the Hamilton Monte-Carlo method, authors have used a simple normalized prior based on the assumption all the brain regions have the same prior distribution (the prior assumption is that all regions are healthy). Can they use a different prior to estimate posterior probability density and likelihood. All the brain areas/tissues are not

healthy. That is the basic presumption and starting point of seizure-like events. Therefore, how a normative prior help in accurately identifying the Posterior distribution of EVs (higher value indicating higher likelihood for seizure) for 8 selected regions obtained from the HMC sampling when analyzing simultaneously SEEG and scalp-EEG above chance level is surprising.

Even though previous studies reported epileptogenic values, and it is routine in clinics to estimate such numbers, I feel the authors should still mention how these values are calculated. They show multiple plots for the posterior distribution of these values, based on which high chance for seizure locations are identified.

Is there any clinical ground truth in this approach (ongoing studies mentioned in the discussion section) that at least tells us the predicted location or site of seizure onset matched with what accuracy for the virtual brain simulation-based predictions? Secondly, what is the ground truth or deviation regarding clinically observed brain network Seizure dynamics in those two patients versus those simulated using the Epileptor-Stimulation model through the TI stimulation? This aspect of this study is not very clear. I am wondering whether it is appropriate to calculate a loss function or gradient descent to see whether the stimulation application on the real brain and virtual brain twin is nearly optimized.

Reviewer #3 (Remarks to the Author):

This paper describes the use of personalised whole-brain neural models in epilepsy. The approach is based on neural mass models of brain voxels segmented from a patient's MRI with conductivity informed by DTI. Parameters such as epileptogenicity and global network scaling of each node are optimised using Hamiltonian Monte Carlo (HMC) with Bayesian inference methods.

The main approach was described in an earlier publication (Wang et al., Sci Trans Med 2023). The authors expand on the earlier work by introducing stimulation-evoked epileptic activity. They tested invasive stimulation via SEEG and noninvasive stimulation via temporal interference (TI).

They first present simulated results from one female patient with occipital lobe epilepsy, including the use of three spontaneous seizures to construct the brain neural mass model and then to stimulate the effect of invasive stimulation via SEEG electrodes and non-invasive TI stimulation. In each stimulation case, the epileptogenic zone network (EZN) was estimated and compared to the known O2 epileptic zone of the patient. The stimulating field was modelled via FEM. They then repeat the simulated pipeline on another patient's data.

Overall, it is important work that can augment the surgical management of such epileptic patients and even guide neuromodulation therapies.

The main limitation of the paper is that the simulations of SEEG/TI stimulation were not experimentally validated (i.e., validation was only done by comparing the simulated discovered EZN with the real EZN)).

The paper can be potentially improved by

- If SEEG recording under stimulation is available for the patients, compare the simulation to the recording.
- Provide insights into the dependency of the seizure activity and/or network span on the stimulation parameters (e.g., frequency and amp).
- Compare the amplitude and distribution of the EZN activation under SEEG and Ti stimulations.

Version 1:

Decision Letter:

Our ref: NATCOMPUTSCI-24-1675A

14th April 2025

Dear Dr. Wang,

Thank you for submitting your revised manuscript "Virtual brain twins for stimulation in epilepsy" (NATCOMPUTSCI-24-1675A). It has now been seen by the original referees and their comments are below. The reviewers find that the paper has improved in revision, and therefore we'll be happy in principle to publish it in Nature Computational Science, pending minor revisions to satisfy the referees' final requests and to comply with our editorial and formatting guidelines.

TRANSPARENT PEER REVIEW

Nature Computational Science offers a transparent peer review option for original research manuscripts. We encourage increased transparency in peer review by publishing the reviewer comments, author rebuttal letters and editorial decision letters if the authors agree. Such peer review material is made available as a supplementary peer review file. **Please remember to choose, using the manuscript system, whether or not you want to participate in transparent peer review.**

Please note: we allow redactions to authors' rebuttal and reviewer comments in the interest of confidentiality. If you are concerned about the release of confidential data, please let us know specifically what information you would like to have

removed. Please note that we cannot incorporate redactions for any other reasons. Reviewer names will be published in the peer review files if the reviewer signed the comments to authors, or if reviewers explicitly agree to release their name. For more information, please refer to our [FAQ page](https://www.nature.com/documents/nr-transparent-peer-review.pdf).

Thank you again for your interest in Nature Computational Science. Please do not hesitate to contact me if you have any questions.

Sincerely,

Ananya Rastogi, PhD
Senior Editor
Nature Computational Science

ORCID

Reviewer #1 (Remarks to the Author):

I believe my concerns were adequately addressed. I would like to congratulate the authors for a nice study.

Arpan Banerjee

Reviewer #1 (Remarks on code availability):

I have reviewed the codes and the readme files and rest of the files are in a replicable format.

Reviewer #2 (Remarks to the Author):

Thank you for carrying out an excellent revision to address all the points. I have no further comments.

Reviewer #2 (Remarks on code availability):

I have checked out their Python implementation in detail. I ran some of their scripts. Overall, the code directory is excellently organized and meets my expectations. I think any external users should be able to run these codes easily with some help.

Reviewer #3 (Remarks to the Author):

The authors sufficiently addressed the reviewers' comments, thereby enhancing the manuscript's clarity and utility. I appreciate their efforts and cooperation.

I have no further comments and am excited about their ongoing experimental validation studies.

Nir Grossman

Version 2:

Decision Letter:

Dear Dr Wang,

We are pleased to inform you that your Article "Virtual brain twins for stimulation in epilepsy" has now been accepted for publication in Nature Computational Science.

Once your manuscript is typeset, you will receive an email with a link to choose the appropriate publishing options for your paper and our Author Services team will be in touch regarding any additional information that may be required.

Authors may need to take specific actions to achieve [compliance](https://www.springernature.com/gp/open-research/funding/policy-compliance-faqs) with funder and institutional open access mandates. If your research is supported by a funder that requires immediate open access (e.g. according to [Plan S principles](https://www.springernature.com/gp/open-research/plan-s-compliance)) then you should select the gold OA route, and we will direct you to the compliant route where possible. For authors selecting the subscription publication route, the journal's standard licensing terms will need to be accepted, including [Plan S principles](https://www.springernature.com/gp/open-research/plan-s-compliance)

[href="https://www.springernature.com/gp/open-research/policies/journal-policies">self-archiving policies](https://www.springernature.com/gp/open-research/policies/journal-policies). Those licensing terms will supersede any other terms that the author or any third party may assert apply to any version of the manuscript.

Acceptance of your manuscript is conditional on all authors' agreement with our publication policies (see <https://www.nature.com/natcomputsci/for-authors>). In particular your manuscript must not be published elsewhere and there must be no announcement of the work to any media outlet until the publication date (the day on which it is uploaded onto our web site).

Before your manuscript is typeset, we will edit the text to ensure it is intelligible to our wide readership and conforms to house style. We look particularly carefully at the titles of all papers to ensure that they are relatively brief and understandable.

Once your manuscript is typeset, you will receive a link to your electronic proof via email with a request to make any corrections within 48 hours. If, when you receive your proof, you cannot meet this deadline, please inform us at rjsproduction@springernature.com immediately.

If you have queries at any point during the production process then please contact the production team at rjsproduction@springernature.com.

We welcome the submission of potential cover material (including a short caption of around 40 words) related to your manuscript; suggestions should be sent to Nature Computational Science as electronic files (the image should be 300 dpi at 210 x 297 mm in either TIFF or JPEG format). We also welcome suggestions for the Hero Image, which appears at the top of our [home page](http://www.nature.com/natcomputsci); these should be 72 dpi at 1400 x 400 pixels in JPEG format. Please note that such pictures should be selected more for their aesthetic appeal than for their scientific content, and that colour images work better than black and white or grayscale images. Please do not try to design a cover with the Nature Computational Science logo etc., and please do not submit composites of images related to your work. I am sure you will understand that we cannot make any promise as to whether any of your suggestions might be selected for the cover of the journal.

Best regards,

Ananya Rastogi, PhD
Senior Editor
Nature Computational Science

P.S. Click on the following link if you would like to recommend Nature Computational Science to your librarian: <https://www.springernature.com/gp/librarians/recommend-to-your-library>

** Visit the Springer Nature Editorial and Publishing website at <http://editorial-jobs.springernature.com> for more information about our career opportunities. If you have any questions please click <http://editorial-jobs.springernature.com>

href="mailto:editorial.publishing.jobs@springernature.com">here.**

Notes on revision made to Nature Computational Science manuscript NATCOMPUTSCI-24-1675

Response to the reviewers

Reviewer 1

Reviewer Comment 1.1 — General: Interventions for drug resistant focal epilepsy rely on the identification of the Epileptogenic Zone (EZ). The current approach involves using stereo EEG (sEEG) electrodes to record seizure activity. A complementary method is to induce seizures via electrical stimulation, either through sEEG electrodes, or via Temporal Interference (TI).

The authors have built on advances in whole brain modeling, individualized parameter estimation and phenomenological models of epilepsy to construct a virtual analog to this pipeline. Patient MRI and sEEG data are used to determine the surface geometry, long range connectivity, epileptogenicity and global network scaling. These are used as inputs to a neural mass model, with node dynamics being determined by the epileptor model, adapted for stimulation. Stimulus is given to the model via an sEEG to source mapping, and the resulting source activity is mapped onto a participant specific cortical surface. Bayesian inference is then applied to determine the EZ.

The authors present two case studies on whom the novel pipeline has been applied. They report that the EZ determined by the model corresponds to the areas whose removal via resective surgery lead to a reduction in epileptic seizures.

Overall, the work is of very high importance and impact. Nonetheless, I also find some serious issues that need addressal.

Reply: Thank you very much for your thoughtful, positive, and encouraging feedback. Please find below our responses to each of your comments.

1 Major Issues: Methodological

Reviewer Comment 1.2 —

Activity is simulated at each vertex (20284), however heterogenous coupling is only used for connections between regions in the VEP parcellation (162). The local coupling is distance dependent and is scaled by a factor γ_{hom} which is invariant across ROIs. Considering that increased connectivity within the EZ is observed in focal epilepsy, homogenous local coupling may be a strong assumption.

Reply: Thank you so much for this valuable and helpful comment. Heterogeneous local coupling indeed plays a critical role in brain networks, as also noted by the reviewer in Comment 1.5, where we will address it systematically. Here, we will focus specifically on the increased connectivity within the EZ in focal epilepsy. In the revised version of this paper, we gave a use case demonstrating the impact of increased local connectivity within the EZ network under SEEG stimulation (in Response Fig. 1).

We added the following paragraph in subsection 2.5 validation of the pipeline. We next demonstrated the role of heterogeneity of local connectivity. Here, local connectivity refers to how neuronal

populations at different vertices are connected within a specific area or neighborhood. We introduced such heterogeneity in local connectivity under the hypothesis that within the EZN local connectivity increases due to synaptic remodeling [1] or the loss of inhibitory synapses [2]. Response Fig. 1 illustrates both source-level and sensor-level SEEG and EEG signals when local connectivity within EZN is 2-5 times higher than in other brain regions. Compared to results obtained with homogeneous local connectivity (Fig. 6), only minor differences were observed during seizure periods. However, the neighborhood of the EZ exhibited a lower mean distribution of epileptogenic values from model inference, making it easier to identify the EZ.

Response Figure (1): The effect of increasing local connectivity within EZ networks under SEEG stimulation as shown in Figure 6. (A) Snapshot of source activity (corresponding videos available in the supplementary materials) at two specified time instants for comparison when the local connectivity within EZ γ_{lc}^{EZ} is r times the local connectivity γ_{lc} of other brain regions with $r = 2, 3, 4, 5$. (B) When $\gamma_{lc}^{EZ} = 2\gamma_{lc}$, selected simulated SEEG time-series (top) and EEG time-series (middle) from SEEG stimulation-induced seizure. The scaled-up time series during the seizure period is shown in red. Posterior distribution of EVs (bottom) from the HMC sampling are shown when analyzing simultaneously SEEG and scalp-EEG. (C) Same as B, but under the condition $\gamma_{lc}^{EZ} = 5\gamma_{lc}$.

Reviewer Comment 1.3 — The site of stimulation used in the virtual pipeline was determined based on the regions resected during surgery, and appears to be proximal to the previously identified

EZ. It is possible that the model is producing type 1 errors by identifying most stimulated regions as epileptogenic. To rule out this possibility, I suggest delivering the same stimulus to an unrelated part of the cortex, and observing if the model still identifies it as an EZ.

Reply:

Thank you so much for this thoughtful and pertinent comment. In a clinical setting, clinicians aim to avoid both false negative and false positive stimulation results (type 1 errors, i.e. non-habitual seizures being triggered) [3–5]. In our study, we avoided false positive seizures induced by SEEG stimulation by a carefully designed protocol. Thanks to the reviewer’s suggestion, we systematically evaluated the effects of stimulation in two settings: 1) stimulating all pairs of electrode leads within the same electrode (PM’) using the same stimulation parameters, as shown in Response Fig. 2. 2) Stimulating 10 additional SEEG bipolar pairs implanted in 10 different cortical regions, as shown in Response Fig. 3.

In summary, our simulations demonstrated that stimulation does not induce seizures in healthy regions. The theoretical basis for this lies in the seizure threshold of each region, which relates directly to its epileptogenicity. The seizure threshold in EZN is lower than in healthy regions. This parametrization ensures that weak stimuli – based on clinically applied stimulation parameters – can trigger seizures in EZN but not in healthy regions, as shown in our model. We also added the following paragraph in subsection 2.5 validation of the pipeline.

Another application of the proposed pipeline is to test and mitigate false positive stimulation results, (i.e., non-habitual seizures being triggered) through a carefully designed protocol and optimized stimulation and stimulation parameters. Here, we systematically evaluated the effects of stimulation in terms of stimulation locations in two settings: 1) Within-electrode stimulation (PM’): Stimulating all pairs of electrode leads within the same electrode using identical stimulation parameters (see Response Fig. 2). 2) Distributed cortical stimulation: Stimulating 10 additional SEEG bipolar pairs implanted in 10 different cortical regions (see Response Fig. 3). Our simulations demonstrated that stimulation does not induce seizures in healthy regions. The theoretical basis for this lies in the seizure threshold of each region, which is directly related to its epileptogenicity. The epileptogenic zone (EZN) has a lower seizure threshold compared to healthy regions. This parameterization ensures that weak stimuli—based on clinically applied stimulation parameters—can trigger seizures in the EZN but not in healthy tissue.

Response Figure (2): Stimulation location was varied alongside the PM' electrode, while all other stimulation parameters remained constant. Fourteen cases are represented. Top Left: the stimulation field for each of the corresponding stimulation locations. Top Right: the resulting brain activity shortly after stimulation, measured at the time point marked in blue in the time series (bottom). The stimulation effects, applied from $t = 500$ ms to $t = 3500$ ms, are visible in all cases but vary depending on the stimulation site. In six out of fourteen electrode pairs, a seizure was triggered in the epileptogenic region 'Left-T1-lateral-prefrontal'. In these cases, the stimulating electrodes were sufficiently close to this area to trigger a seizure. In all other cases, the stimulating electrodes were too far from the epileptogenic zone (EZ) to trigger a seizure in the regions where they generated a focal electric field, due to those regions having a higher seizure threshold than the EZ.

Varying stimulation location along random electrodes

Response Figure (3): Stimulation location varied across different electrodes, while all other stimulation parameters remained constant. Ten cases are represented. In each case: Top Left—stimulation electrode highlighted in blue and stimulating pair shown in red and orange. Top Right—estimated stimulation field from the corresponding stimulation location. Bottom—simulated time series of brain activity before, during and after stimulation. The stimulation effects, applied from $t = 500$ ms to $t = 3500$ ms, are visible in all cases but vary depending on the stimulation site. Different non-epileptogenic brain regions were targeted by the focal stimulation field and no seizure was triggered.

2 Major Issues: Future Directions

Reviewer Comment 1.4 —

Can the Epileptor model parameters be accurately estimated from surface EEG alone? If so, this method coupled with Temporal Interference could provide an entirely non-invasive treatment for drug resistant epilepsy.

Reply: We fully agree with your comment here. Thank you. For the example patient, we did obtain a good estimate of the EZN which was located in cortical regions. Based on this study, we are currently systematically investigating the estimation of the EZN from scalp-EEG signals within a cohort of patients. We are testing the robustness of the VEP with regard to variability in types of epilepsy, as well as spatial location of the EZ. We furthermore investigate which spatial resolution of the EEG recording device is sufficient to discover the underlying EZN estimation.

Thanks to your suggestion, we revised and extended our current discussion in the discussion section. Below is an updated version.

To achieve non-invasive diagnosis and treatment with temporal interference in the future, we need to optimize the workflow using purely non-invasive data. A key next step, directly building on this work, is to systematically estimate the EZ from scalp-EEG signals within a cohort of patients. In the presented example, we obtained a good estimate of the EZN from non-invasive scalp-EEG signals, with localization in superficial cortical gray matter. Future studies should assess the robustness of the VEP across different epilepsy types and EZ locations. Additionally, the required spatial resolution (i.e., number of scalp-EEG recording channels) for accurately identifying the underlying EZN needs further investigation. Another future direction is to develop a non-invasive VEP using high-resolution MEG data. We have obtained preliminary results with ^{23}Na MRI [6] and now need to develop methods for VEP estimation and reconstruction from high-resolution EEG and MEG data. A further question is how precisely the EZN can be identified using interictal spikes or network features from EEG and MEG — potentially allowing seizure-inducing stimulation to be avoided.

Reviewer Comment 1.5 —

Allowing γ_{hom} to vary across ROIs and determining its values for each participant could add another dimension to the subject specificity of the treatment and improve the method's accuracy.

Reply: Thank you for such a constructive comment. Your comment encouraged us to demonstrate it in a subsection of the section Results (see response to Comment Comment 1.4) and also develop a paragraph in the discussion section below. In the new version of the paper, we updated the notation γ_{hom} to γ_{lc} .

We extended this pipeline by incorporating the capability to model heterogeneous local connectivity in a brain region-specific manner. Three key elements must be taken into account: (1) the specific EZN associated with different types of epilepsy, (2) the type of neural models applied, and (3) personalization. In virtual brain twins, local connectivity should be derived from the anatomical information, rather than from functional connectivity and effective connectivity [7]. Anatomical connectivity varies across different EZNs depending on the type of epilepsy. For example, increased local connectivity within the EZ may result from enhanced excitatory synapse formation driven by axonal sprouting [8, 9] or from the loss of inhibitory synapses [2]. Conversely, neuronal loss, axonal damage, and synaptic pruning may reduce anatomical connectivity, though the exact mechanisms remain unclear [10]. The choice of the neural activity model also plays an important role in representing the heterogeneity of local connectivity. Our results indicate that incorporating local connectivity heterogeneity does not significantly alter brain activity or seizure propagation when using the Epileptor model in this pipeline. The Epileptor, as a phenomenological model, and its excitability parameter x_0 already account for the coexistence of increased excitatory connectivity and decreased inhibitory connectivity, which reflects the hyperexcitability of epileptogenic tissue. However, region-specific local connectivity may play a more significant role in biophysical models [11, 12]. Currently, personalized measurements of local connectivity are not feasible in routine clinical practice. However, advancements in neuroimaging techniques may provide such information for research purposes and, in the future, for clinical applications [13].

3 Minor Issues: Formatting

Reviewer Comment 1.6 —

1. Figure 1: The plots within sub figures B, E, F have axis labels and ticks that are illegible at this scale and create visual clutter.

Reply: Thank you for this comment, which helped us improve Figure 1 for a better clarity. We removed unnecessary and illegible labels and ticks, and added key messages.

Reviewer Comment 1.7 —

2. Line 114; Local and global propagation of seizures across vertices is mentioned separately, ostensibly because the local propagation occurs under homogeneous coupling. However, since that distinction is only made later in the article the phrasing here seems awkward.

Reply: Thank you for your comment on heterogeneous local connectivity. They helped us realize that emphasizing homogeneous local connectivity may be misleading. Here, we described local and global connectivity separately because they have different biophysical meanings and are derived through different methods. Local connectivity primarily arises from synaptic connections and the propagation of fields through space, whereas global connectivity is mediated by white matter fiber connections.

Reviewer Comment 1.8 — 3. Line 161; Consider replacing “with the hypothesis” with “to test the hypothesis”.

Reply: Thank you for this comment. Done.

Reviewer Comment 1.9 — 4. Figure 7: In the caption, the heading for subfigure E is not bold, as the other headings are.

Reply: Thank you for this comment. Done.

Reviewer Comment 1.10 —

Codes are fine. readme files are in place for guidance that are adequately commented

Reply: Thank you so much for your positive confirmation.

Reviewer 2

Reviewer Comment 2.1 — The present study attempts to use the virtual brain twin concept based on their previous work using VEP models and frameworks. The Virtual brain twins are personalized, generative, and adaptive brain models based on data from an individual patient’s brain for scientific and clinical use. In this study, they introduce the first high-resolution virtual brain twin workflow, specifically designed for estimating the EZN involving direct electrical stimulation through SEEG electrodes. Overall, I think this work is important and has the potential to advance individual-specific brain mapping and identification of EZN estimation under a variety of invasive and non-invasive clinical settings. However, I have a few fundamental queries related to workflow implementations and the translation from invasive to non-invasive settings. Some more major and minor comments are outlined below. I sincerely hope that addressing these may improve the clarity of the methods and presentation of the findings in the revised version of this manuscript.

Reply: Thank you for your positive general comment. Yes, we do believe that addressing your comments below will enhance the clarity of the methods and improve the presentation of the findings in the revised version. Thank you again.

Reviewer Comment 2.2 — In the data processing step to construct individual brain network models, the authors used parcellation of the cortex using a VEP atlas. Are there any limitations in using other atlas for the individual connectome construction? It is perhaps not clear what the scientific rationales are for choosing the VEP atlas for the individual connectome of the patients over other existing atlases.

Reply: Thank you for this valuable comment. The choice of atlas is indeed a crucial decision in a scientific study. The pipeline proposed in this paper can be easily adapted to other atlases for individual connectome construction as well as in the model inversion modules. We have provided examples in our response to Comment 2.4. In the updated version, we have also added the scientific rationale for choosing the VEP atlas.

The reasons for choosing the VEP atlas are as follows: 1) The VEP atlas is specifically designed for epileptology, incorporating anatomical and functional features of each brain structure, particularly in relation to the EZN and surgical applications. 2) The geometric features and sizes of the brain regions are well suited for both clinical applications and modeling, including model inversion. 3) Brain regions can be automatically labeled and personalized from T1-MRI scans using geometric and neuroanatomical information. 4) The VEP Atlas has been clinically evaluated in a retrospective study including 53 patients, and is currently being assessed in another prospective trial with 356 patients, ensuring its clinical suitability and reliability [14, 15].

Reviewer Comment 2.3 — VEP atlas is derived from their previous study the code/script and reference are included. However, I feel some more details should be provided in this manuscript to aid the readers in understanding the implemented pipeline.

Reply: Thank you for the suggestion. In the updated version, we have provided a summary of the implemented pipeline for the VEP atlas as follows. A brief summary of the steps to obtain the VEP atlas goes as follows. T1-weighted images are processed using FreeSurfer to remove non-brain tissue [16], segment subcortical gray matter structures [17], normalize intensity [18], and generate cortical surfaces [19]. These surfaces are inflated, registered to a spherical template, and corrected for topology [17, 20]. The cortex is then subdivided into regions based on gyral and sulcal structures [21], forming the basis for constructing the VEP atlas. This construction involves splitting, merging, and renaming operations to both cortical and subcortical regions. Cortical regions are divided based on the triangulated surface mesh, while subcortical regions are split directly on voxels. Nonlinear splits are applied in specific areas with high curvature, such as the callosal sulcus, whereas the superior frontal gyrus is split using a combination of linear and specialized methods based on cortical surface geometry [22].

Reviewer Comment 2.4 — Are the results of the stimulation sensitive to the choice of the VEP atlas and location of the brain regions determined by the atlas, or is it generalizable and the stimulation results based on SEEG/Scalp EEG data are reproducible regardless of the choice of this atlas to construct virtual brain twins? Perhaps the authors can explain this in more detail.

Reply: Thank you for this great comment and suggestion. In theory, within the high-resolution simulation pipeline, the results of stimulation are not highly sensitive to the choice of the atlas. The key differences arise from the size and location of EZNs, as well as structural connectivity. However, the estimation of EZNs by model inversion may vary due to differences in parcellation across different atlases.

In this revision, we extended the entire pipeline to adapt to different atlases. We also provided an example using the Desikan-Killiany (DK) atlas, where the EZN network (superior frontal gyrus) is much larger than in the VEP atlas. In the VEP atlas, the superior frontal gyrus is divided into five distinct structures. We applied the same stimulation effect, which is defined at the vertex level and is independent of the atlas. In this extreme case, we observed that while the simulation effects and SEEG/scalp EEG data remain similar, the model inversion for diagnosis differs significantly. A detailed description and integration are provided in the following paragraph in the updated version of the paper.

The choice of atlas is an important topic in neuroscience studies, specifically for whole-brain network studies, and should be aligned with the specific research objectives. We selected the VEP atlas [22] because it is purposefully designed for epileptology, considering both anatomical and functional features within each structure, particularly in relation to EZN and surgical applications. Moreover, we also extended our pipeline to be easily adapt to other atlases, such as Desikan-Killiany (DK) atlas [23,24] or Schaefer Atlas [25]. We demonstrated this application using the DK atlas while maintaining the same simulation parameters as those used for the VEP atlas, including the stimulation electric field in supplementary Fig. 4. The simulation of source activity and SEEG/scalp EEG remains similar, particularly in terms of spatiotemporal patterns. This is because the primary difference arises from variations in the size of the EZN and structural connectivity (See Fig. 4 A-C). The forward solution remains unchanged due to the high-resolution stimulation at the source level. However, the estimated EZN is significantly larger in the DK atlas because its brain regions are more extensive compared to those in the VEP atlas. As an example, we examined a case where the EZN network in the superior frontal gyrus is substantially larger in the DK atlas than in the VEP atlas. In the VEP atlas, the superior frontal gyrus is subdivided into five distinct structures.

Response Figure (4): Simulating and estimating the EZN in the pipeline using the Desikan-Killiany (DK) atlas. **(A)** Snapshot of source activity (corresponding videos available in the supplementary materials) at four specified time instants under SEEG stimulation using DK atlas. **(B)** Selected simulated SEEG time-series and **(C)** Scalp EEG time-series from SEEG stimulation-induced seizure. The scaled-up time series during the seizure period is shown in red. **(D)** Posterior distribution of EVs from HMC sampling when analyzing simultaneously SEEG and scalp-EEG using the DK atlas. **(E)** Heatmap of the estimated EZN in T1-MRI from the pipeline using the DK atlas, based on results from **(D)**. **(F)** Heatmap of estimated EZN in T1-MRI from the pipeline using the VEP atlas, based on results from Figure 6 **(D)**.

Reviewer Comment 2.5 — The authors mentioned that exposure to repetitive stimulation/perturbation generates a slow accumulation effect, which can push the system to the seizure state when a seizure threshold is reached. Beyond this excitability threshold, a network of neurons likely exhibits a high firing rate and increase in spiking activity.

The authors use this logic to introduce a slow variable that mimics the accumulation effect in the mathematical model. Mathematically, the time-dependent accumulation variable $m(t)$ in the model is well described and acts like a switch regarding perturbations introduced by brain stimulations. However, $m(t)$ and its relationship with a fast-spiking variable need to be defined better, and the link of $m(t)$ to underlying biophysics should be better described. Additionally, we know that perturbation effects are very specific and depend on various factors, the magnitude of the current

pulse, pulse width, electric field distributions, and stimulation frequency tailored to the subject's response to field oscillations, Excitation-Inhibition balance (Gaba-Glutamate ratio). Those factors have substantially differential effects on the excitability of a network of spiking neurons. Hence, in their phenomenological model, what exactly accumulates as represented by the slow variable here is a bit fuzzy. How these accumulation time scales relate to the dynamics of fast variables could be important for understanding the entry to the seizure state once an appropriate threshold is reached. I find this in the current version of the implementation, and the description is unclear.

Reply:

Thank you for pointing out several important questions related to $m(t)$. To address the first point regarding $m(t)$ and its relationship with the fast-spiking variables, we have conducted phase plot and bifurcation analysis by varying m in the Epileptor model. We have added the necessary explanations in the Methods section as follows. Since $m(t)$ plays an important role in the model with stimulation, Supplementary Fig. 5 demonstrates how the system behavior changes as m varies. As m increases from 0 to 1.5, which is below the threshold ($m_{thresh} = 1.8$), the oscillation in the upstate changes from stable spiral to a spiral with limit cycles (see Supplementary Fig. 5 **A-C**). Higher values of m correspond to larger ranges of limit cycles in both dimensions, x and z . When $m = 2.0$, which exceeds $m_{thresh} = 1.8$, m directly influences z , pushing the system into the seizing state.

Second, to address the link between $m(t)$ to underlying biophysics, we have included the following sentences in the updated version of the paper: The variable $m(t)$ is directly related to the excitability of the corresponding tissues. Tissue excitability in epileptogenic networks arises from a combination of factors, including ion channel dysfunction, an imbalance between excitation and inhibition, and altered ion homeostasis. However, care must be taken not to overinterpret its role, as we are working with a phenomenological model.

Third, regarding the question of what exactly accumulates as represented by the slow variables, our response is included in the revised version of the paper: There are two slow variables, $z(t)$ and $m(t)$. While z drives the gradual changes underlying seizure initiation and termination, m increases by accumulating the stimulation effects through the influence of the electric field. As m increases, the stability of the system changes. Once m reaches the threshold value $m_{threshold}$, it directly pushes z into a state that prepares the system to transition into a seizure state. In addition, in another paper [26], we demonstrated patient-specific stimulation in terms of magnitude of the current pulse, stimulation frequency and stimulation location.

Response Figure (5): Varying m in a single Epileptor model. The 3D Epileptor model with varying m . Each row, from top to bottom: time series for each variable (x_1 , y_1 , z), phase plane plot with nullclines x_1 and y_1 for $z = 3.1$ and a complete bifurcation diagram for varying values of z (dotted line: unstable fixed points, continuous line: stable fixed points). A) $m = 0$, the oscillations in the upstate are brief because the equilibrium in the upstate is a stable spiral. B) $m = 0.5$, oscillations in the upstate are of higher amplitude and longer because the equilibrium in the upstate is an unstable spiral with a limit cycle (in red in the phase plane plot). C) $m = 1.5$, oscillations in the upstate have an even higher amplitude and last for the entire duration of the upstate, the limit cycle, shown in red in the phase plane has a greater diameter. D) $m = 2$, the system exhibits unstable spiral behaviour.

Reviewer Comment 2.6 — The author mentions local connections are heterogeneous; however, to include that in the virtual brain twin model is nearly impossible at this stage as the pieces of information needed on interneurons and dendritic arborization in clinical measurements have yet to be made available. However, they only consider geometric features of cortical surfaces for the epileptic simulations. It is again unclear what geometric features of cortical surface they are talking about. Do they mean dipole orientation and magnitude scaling as a function of the underlying source location that goes into LED field matrix calculations for SEEG/EEG stimulations? As I understand, they are only using an individual connectome, as shown in the workflow diagram in Figure 1, which incorporates streamlined counts and fibre orientation density measured from DW-MRI.

Reply: Thank you for your comments. We agree that the creation of a virtual brain twin model of personalized local connectivity is nearly impossible at this stage, as the necessary clinical measurements are not yet available. We have formulated this information into a paragraph in the discussion section.

Here, we clarify the points that were unclear. The geometric features of cortical surfaces are derived from T1-MRI, where the vertices are obtained from the patient's specific gyri and sulci shapes. The

whole-brain simulation is based on these vertices (with geometric features), which is further constrained by local connectivity, global connectivity (individual connectome), and the forward solution. The strength of local connectivity is defined as a function of the distance between the vertices. The location of the vertices is related to the definition of individual brain regions, which, in turn, are related to the global connectivity.

Regarding the forward solution, the geometric features can influence dipole orientation and magnitude scaling, depending on the underlying source location, which is used in the lead field matrix calculation for SEEG/EEG simulations.

If the reviewer permits further discussion, the virtual brain twins include two levels of personalization. The first-level of personalization involves individual structural information based on anatomical imaging such as T1-MRI, DWI-MRI, and CT scans. This structural information includes the geometric features of cortical surfaces, subcortical volumes, and connectome data. The second-level of personalization includes key parameters inferred from functional data, such as SEEG and EEG.

Reviewer Comment 2.7 — The author says we map the stimulus signal onto the parcellated brain areas based on distance. What stimulus they are describing needs to be mentioned. Perhaps that information is important for the readers.

Reply: Thank you for your helpful comment, which encouraged us to revise the subsection 'Calculation of the Electric Field of SEEG Stimulation' in the Methods and Materials section. We hope this revised version is now clear.

The stimulus signal, ψ , is a waveform represented here as a biphasic pulse train of electrical current. The electric field strength at each vertex depends on the distance between vertices and the stimulated pair of bipolar electrode leads. Shorter distances result in higher field strengths. To simplify, we identified the maximal field strength within each brain region and uniformly applied it to all vertices within that region, as shown at the bottom of Fig. 3 **A** and Fig. 6 **A**. We defined the I_{stim} at each vertex as $\psi \times$ electric field strength at this vertex, with an example shown in supplementary Fig. A2. Please note that the effects of the stimuli on brain signals are dependent on multiple factors, including the nonlinearity of local dynamics, local and global connections, and the stimulation current.

Reviewer Comment 2.8 — This is related to the earlier question as well. If I understand their pipeline, the authors have created a virtual brain twin based on the connectivity estimate from patients' diffusion imaging data. Once they have the virtual brain twin of the individual patient, they will apply a relatively new tACS stimulation technique where vectorial tACS fields are added together, resulting in a T1 field estimate for the brain regions. Subsequently, they used this estimated field as an input to the I_{stim} parameter and simulated their phenomenological model on one node/vertex on the connectome (from the identified EZ). However, my question is whether the stimulation protocol or, in other words, the T1 field needs to be tailored in an individual/patient-specific manner or whether the individual specificity only lies in the virtual brain twin constructed in the pipeline based on the individual connectome. My conceptual understanding and assumption are that individual connectome requires individual stimulation protocol likely for the most effective clinical outcome of the adjuvant treatment strategy. Please explain what I am missing here.

Reply: Thank you for this important comment. We fully agree that an individual stimulation protocol is necessary for achieving the effective clinical outcome of the adjuvant treatment strategy. Our pipeline is designed to incorporate an individualized stimulation protocol for each patient. In this paper, the virtual

brain twins incorporate three-level of personalization. 1. First-level: Individual geometric information and connectome derived from anatomical data such as MRI. 2. Second level: Estimation of key parameters, such as spatial distribution of excitability (related to EZN), derived from functional data such as SEEG and EEG. 3. Third-level: personalized stimulation protocol. The electric field calculations are based on individual MRI data, taking into account brain geometry and head tissue segmentation. The personalized spatial distribution of excitability provides a tailored response to the stimulation protocol, enhancing the effectiveness of the intervention.

We have added the following paragraph to subsection 3.1 in the discussion section. Thank you again for your valuable comment. The virtual brain twins for stimulation incorporate three levels of personalization, adding one more level compared to previous versions [14, 27]. These include: (1) individual geometry and connectome derived from anatomical data such as MRI, (2) estimation of key parameters, such as spatial distribution of excitability (related to EZN), from functional data such as SEEG and EEG, (3) personalized stimulation protocol. The electric field calculations are based on individual MRI data, taking into account brain geometry and head tissue segmentation. The personalized spatial distribution of excitability provides a tailored response to the stimulation protocol, enhancing the effectiveness of the intervention. "

Reviewer Comment 2.9 — For the model inversion using the Hamilton Monte-Carlo method, authors have used a simple normalized prior based on the assumption all the brain regions have the same prior distribution (the prior assumption is that all regions are healthy). Can they use a different prior to estimate posterior probability density and likelihood. All the brain areas/tissues are not healthy. That is the basic presumption and starting point of seizure-like events. Therefore, how a normative prior help in accurately identifying the Posterior distribution of EVs (higher value indicating higher likelihood for seizure) for 8 selected regions obtained from the HMC sampling when analyzing simultaneously SEEG and scalp-EEG above chance level is surprising.

Reply: Thank you for pointing out this topic regarding the prior, which encouraged us to add a paragraph in the discussion section. The informative prior indeed helps to obtain the posterior distribution more efficiently, as demonstrated in our previous work. The noninformative prior is more challenging. After improving the model inversion modules, the algorithm is still able to identify the targeted EZN even with a noninformative prior. Then we decided to publish the results with noninformative prior. This effectiveness can be attributed to the current model inversion algorithm, which incorporates reparameterization and combines both maximum a posteriori (MAP) estimation and the HMC algorithm. More detailed information can be found in the methods section, and the shared codes are available on Code Ocean. The informative prior plays an important role in the calculations of the posterior, enabling reliable and efficient evaluation of potential hypotheses [28]. In the EPINOV clinical trial, we used the informative prior derived from the data analysis of SEEG onset features [14] in maximum a posteriori (MAP) estimation algorithm. We also tested whether the brain sodium MRI-derived prior could slightly improve the performance of VEP [6]. In this paper, we further improved our model inversion algorithm, demonstrating that even with an uninformative prior, the algorithm can still identify the targeted EZN. The effectiveness is due to the current model inversion algorithm, which includes reparameterization and a combination of both MAP estimation and the HMC algorithm.

Reviewer Comment 2.10 — Even though previous studies reported epileptogenic values, and it is routine in clinics to estimate such numbers, I feel the authors should still mention how these

values are calculated. They show multiple plots for the posterior distribution of these values, based on which high chance for seizure locations are identified.

Reply: Thank you for this suggestion. We have added the subsection "Calculation of the epileptogenic values" in the Section of "Methods and Materials", as shown below. Using the HMC model inversion algorithm, we obtained the estimated source time series, and based on these, we calculated brain region-specific epileptogenicity values (EVs). We checked the source time series (variable x of the 2D Epileptor in equation (3) of region i for values above a threshold of 0. The first occurrence of such a value is considered to be the onset of the seizure t_i in that region. We define $t_0 = \min(t_i), i = 1, \dots, 162$. Brain regions with no estimated seizure (i.e., no values above 0) are assigned an onset value $t_i = 200$. The EV_i for brain region i is calculated as $EV_i = -\log(((t_i - t_0) + 1)/20)$. Then we normalized the EV vector to $[0, 1]$ for each sample and plotted the distribution of EVs.

Reviewer Comment 2.11 — Is there any clinical ground truth in this approach (ongoing studies mentioned in the discussion section) that at least tells us the predicted location or site of seizure onset matched with what accuracy for the virtual brain simulation-based predictions? Secondly, what is the ground truth or deviation regarding clinically observed brain network Seizure dynamics in those two patients versus those simulated using the Epileptor-Stimulation model through the TI stimulation? This aspect of this study is not very clear. I am wondering whether it is appropriate to calculate a loss function or gradient descent to see whether the stimulation application on the real brain and virtual brain twin is nearly optimized.

Reply: Thank you for this important question. There is no clinical ground truth of the EZN. However, when we designed our cases to run the pipeline, we did consider the following clinical information. For example, in the case of Patient 1, we constructed a virtual brain twin using the patient's structural and functional data. The functional data included SEEG spontaneous ictal signals, and through model inference, we identified the EZN as the left-O2 region. During surgery, half of the left-O2 region was removed, resulting in an Engel class outcome of II, indicating the patient was almost seizure-free, with fewer than three seizure days per year. Therefore, we took left-O2 as the EZN for calibrating key personalized parameters to generate virtual brain twin simulation data. Two factors are related to the comparison between simulated data and empirical data, quantitatively and visually. Two factors guided the comparison between simulated and empirical data: quantitative and visual alignment. During model inversion, which was based on spontaneous seizures, our algorithm utilized fitness functions that compared seizure propagation patterns (both temporal and spatial information) in simulated and empirical data. Visually, compared with Figure 2A and Figure 2C, we can see the similarity of seizure spatial propagation patterns. In the case of Patient 2, we chose the left-F1-lateral-prefrontal region as the EZN network based on two factors of clinical information: (1) the left-F1-lateral-prefrontal region is part of the EZN network identified through VEP estimation on the patient's spontaneous seizure, and (2) it is also within the regions removed during resection surgery, after which the patient has been seizure-free.

Reviewer 3

Reviewer Comment 3.1 —

This paper describes the use of personalised whole-brain neural models in epilepsy. The approach is based on neural mass models of brain voxels segmented from a patient’s MRI with conductivity informed by DTI. Parameters such as epileptogenicity and global network scaling of each node are optimised using Hamiltonian Monte Carlo (HMC) with Bayesian inference methods. The main approach was described in an earlier publication (Wang et al., *Sci Trans Med* 2023). The authors expand on the earlier work by introducing stimulation-evoked epileptic activity. They tested invasive stimulation via SEEG and noninvasive stimulation via temporal interference (TI). They first present simulated results from one female patient with occipital lobe epilepsy, including the use of three spontaneous seizures to construct the brain neural mass model and then to stimulate the effect of invasive stimulation via SEEG electrodes and non-invasive TI stimulation. In each stimulation case, the epileptogenic zone network (EZN) was estimated and compared to the known O2 epileptic zone of the patient. The stimulating field was modelled via FEM. They then repeat the simulated pipeline on another patient’s data. Overall, it is important work that can augment the surgical management of such epileptic patients and even guide neuromodulation therapies. The main limitation of the paper is that the simulations of SEEG/TI stimulation were not experimentally validated (i.e., validation was only done by comparing the simulated discovered EZN with the real EZN)).

Reply: Thank you to the reviewer for providing a very nice summary. We agree that there was no experimental validation in this paper. The primary aim of this work is to provide a methodological and conceptual foundation for a series of ongoing scientific studies and clinical applications. The validation of clinical applications is outlined in Section 3.2, ‘Application.’ To name just a few examples: optimizing stimulation parameters for SEEG-induced seizures or functional mapping, evaluating the accuracy and limitations of noninvasive diagnostics like EEG or MEG, and building personalized whole-brain models for noninvasive TI stimulation.

Based on the results presented in this paper, three PhD theses are currently in progress, and three clinical trials are being prepared. We do believe that a dedicated paper is both necessary and important to provide a comprehensive methodological pipeline and establish a clear conceptual foundation.

Reviewer Comment 3.2 — The paper can be potentially improved by – If SEEG recording under stimulation is available for the patients, compare the simulation to the recording. – Provide insights into the dependency of the seizure activity and/or network span on the stimulation parameters (e.g., frequency and amp).

Reply: Thank you for your comments. One immediate application of this pipeline is optimizing stimulation parameters, enabling clinicians to perform stimulation with predicted effects on patients more effectively. The primary purpose of this paper is to demonstrate the pipeline’s capabilities, rather than reiterating empirical data.

We do agree that validation is necessary and important. We conducted a systematic validation and comparison in another study, titled “Virtual epileptic patient cohorts”, where we analyzed 16 SEEG-stimulated seizures across 14 patients [26]. In that study, we applied the EZ hypothesis derived from spontaneous seizures and modeled electrical stimulation based on clinical parameters. Specifically, the spatial distribution of the electric field was computed using stimulation location and charge, while the stimulus waveform was modeled according to stimulation frequency, pulse width, and total duration. The strength of the electric field was then input into the Epileptor-stimulation model and scaled proportionally to its phenomenological variables. This scaling parameter was determined by comparing empirical signals with simulated ones.

We used the same SEEG stimulation model as in the study by Dollomaja et al. (2024). The key difference in the modeling section lies in the spatial resolution: while the Dollomaja et al. (2024) study employed low-resolution simulations using the 162 region VEP atlas, this paper focuses on high-resolution simulations, which are necessary for the realistic stimulation of TI. In theory, the approach presented here should be more accurate, as it incorporates local connectivity and a more precise forward solution, both of which are critical elements, as demonstrated in prior work [29].

Below, we provide examples of simulated seizures for two patients with different stimulation frequencies (Item 1). This is followed by a systematic evaluation and comparison of the simulated data with empirical data (Item 2). Additionally, we present a group analysis of stimulation locations (Item 3) and stimulation amplitudes (Item 4), along with a detailed example from one patient (Item 5).

1) Simulation of stimulated seizures. In Response Fig. 6A, we show an example of a focal temporal seizure from patient 3, triggered by stimulation of electrodes B2 and B3, as anode and cathode, respectively. The epileptogenic network is estimated from the spontaneous seizure of the same patient using the VEP pipeline. The epileptogenic zones are T2-anterior and hippocampus-anterior of the right hemispheres. The propagation zones are SFS-rostral, amygdala and fusiform gyrus of the right hemisphere. Following the stimulation period, a seizure is triggered and recorded in the stimulating and nearby electrodes. The second example consists of a bilateral temporo-frontal seizure from patient 12, applied using electrodes B3 and B4 (Response Fig. 6B). The epileptogenic network is estimated from the spontaneous seizure of the same patient. The epileptogenic zones are left orbito-frontal-cortex, right F3-pars-opercularis and right occipito-temporal-sulcus. Propagation zones have an extended network including the right hippocampus-anterior where the stimulating electrodes are located. Seizure activity is first observed in the right-temporal-lobe, and later propagates to frontal regions and the contra-lateral hemisphere (B' electrodes located in the left temporal lobe). In both examples, normalized signal power distribution on reconstructed SEEG electrodes display the large network of seizure organization.

Response Figure (6): Stimulated seizures induced by adjacent SEEG electrodes for two patients with different propagation patterns. A) Focal temporal seizure of patient 3, induced by high-frequency stimulation of electrode pairs B2-B3, located in the right hippocampus anterior. Stimulation waveform is a bipolar pulse applied at 50 Hz frequency, 2.2 mA amplitude, with pulse width of 1 millisecond and 5 second duration. B) Bilateral seizure of patient 12, induced by low-frequency stimulation of electrode pairs B3-B4, also located in the right hippocampus anterior. Stimulation waveform is a bipolar pulse applied at 1 Hz frequency, 1 mA amplitude, with pulse width of 2 milliseconds and 19 second duration. In both A) and B), left-side panels display stimulation parameters, and the epileptogenic network estimated from the spontaneous seizure of the same patient. Red, orange and light blue represent EZ, PZ and HZ respectively. The middle panels show empirical and simulated SEEG time series for a few electrodes. Red vertical lines denote the seizure onset and offset, determined by clinicians for the empirical recordings and by our model in the simulated time series. The right-side panels show the normalized signal power distribution for all channels. Color bar represents signal power, where blue and red represent low and high signal power, respectively.

2) Evaluating stimulated seizures. We used three approaches to evaluate synthetic stimulated seizures (Response Fig. 7). Before comparison, we removed the time series corresponding to the stimulus current and only compared the post-stimulus time series. In the first approach, similarly to spontaneous seizures, four metrics were compared against a randomized cohort). The randomized cohort was generated using random EZ hypothesis and contains in total 15 stimulated seizures. Thus, for the same patient, an EZ hypothesis was chosen from a random patient. Then, we applied the same stimulation parameters to simulate seizure dynamics induced by stimulation. We employed a permutation test on the comparative metrics that specifically compared means. The results demonstrate a significantly better performance for the virtual epileptic cohort compared to the randomized cohort, as shown in Response Fig. 7A. However, here the seizure propagation metrics (SP) showed no significant differences between the two cohorts.

Response Figure (7): Comparison among simulated SEEG signals with empirical recordings for stimulation induced seizures. A) Four metrics to quantify the comparison of stimulated SEEG seizure time-series in virtual epileptic cohort (VEC, $N=16$ in blue) and the randomized cohort (RC, $N=15$ in red). Each point in the swarm plot corresponds to one metric comparing one empirical and simulated SEEG pair. Black points in each metric represent the mean value. **** p -value < 0.0001 , *** p -value < 0.001 ; permutation test. B) Performance metrics for varying only stimulation locations in seven patients, measured by the distance from the empirical stimulation locations. We randomly stimulated 10 pairs of electrodes within four main distance groups located outside of the empirical locations. If we define d_e as the distance in cm from the empirical locations, Dist1: $d_e \leq 1$; Dist2: $d_e \in [1, 2]$; Dist3: $d_e \in [2, 3]$; and Dist4: $d_e \geq 3$. Four metrics are used to compare the five distance groups in four box plots for seven patients, with individual data points overlaid (in blue). C) Performance metrics for varying only stimulation amplitude in seven patients. Empirical stimulation amplitude varied from 1.8-2.2 mA. Then we performed simulations using the same stimulation parameters but 2 lower amplitudes (0.5 mA, 1 mA) and 2 higher amplitudes (3 mA, 4 mA). Four metrics are used to compare across the five stimulation amplitude groups. In all cases, results are shown in box plots, overlaid over individual data points. Middle box represents the interquartile range (IQR), with a line at the median. The whiskers extend from the box to the data point lying within 1.5x the IQR. Points past the whiskers are marked as fliers.

3) Stimulation location we investigated the role of two stimulation parameters in inducing seizures: stimulation location and stimulation amplitude. We varied these stimulation parameters and compared the simulated outcome against the empirical stimulated seizure. We varied stimulation location by randomly selecting 10 electrode pairs from each of four distance groups to stimulate for seven patients, shown in Response Fig. 7B. The distance groups are defined by d_e the distance from the empirical stimulation location: Dist1: $d_e \leq 1$ cm; Dist2: $d_e \in [1, 2]$ cm, Dist3: $d_e \in [2, 3]$ cm and Dist4: $d_e \geq 3$ cm. For each patient, stimulation parameters are all the same as their empirical cases, except for the stimulation locations. As stimulation location was selected incrementally further away from the empirical location, the similarity between the simulated and empirical seizure dynamics deteriorated, as observed across our four metrics (see also Response Fig. 9). Both structural connectome and the EZ network configurations determined the stimulated seizure patterns.

4) Varying stimulation amplitude. For varying stimulation amplitude, we could evaluate the capacity of our model in generating seizure dynamics for a particular stimulation amplitude (Response Fig. 7C). For each patient, first, we adjusted the model parameters to induce seizures by stimulation using the same stimulation amplitude as the empirical case. Then, we varied stimulation amplitude using common amplitudes used clinically (two lower and two higher amplitudes than the empirical one) to simulate the signals and evaluate the post-stimulus response by comparing it with the empirical stimulated seizure. Lower stimulation amplitudes did not induce a seizure, which translated to low similarity values across metrics. Higher stimulation amplitudes induced synthetic seizures which were similar to the empirical amplitudes, but longer lasting (see Response Fig. 8).

5) Changing stimulation parameters: patient examples An example from patient 10 of the cohort is shown in Response Fig. 8 where different stimulation amplitudes are tested in-silico and compared against the empirical data. Stimulation is applied in electrodes TB'1-TB'2, located in proximity to the left hippocampus region. For all simulations, the EZ hypothesis is estimated from the VEP pipeline and consists of the following regions: Left-Hippocampus-anterior, Left-Hippocampus-posterior, Left-Amygdala. The accumulation hypothesis embedded in the model is such that the effect of the stimulation on the model depends on the seizure threshold. If the critical threshold for seizure onset is reached, the model is kicked to the seizure state (Response Fig. 8C) otherwise the model stays in its normal state (Response Fig. 8B). Higher stimulation amplitudes will always cross this threshold and destabilise the system to the seizure state (Response Fig. 8D).

For the same patient, an example with different stimulation locations is shown in Response Fig. 9. The stimulation location is chosen by randomly selecting a pair of electrodes within a certain radius from the empirical stimulation location. As the stimulus is applied increasingly further away from the empirical stimulation location, the seizure dynamics progressively change from the empirical post-stimulation response.

Response Figure (8): Simulation examples with different stimulation amplitude from virtualized patient 10. Double arrow indicates stimulation period. Seven channels are plotted in bipolar montage (out of 116 total bipolar channels). Vertical red lines indicate seizure onset and seizure offset. For all plots B), C) and D), upper plots show simulated time series at the SEEG level with stimulation applied at different amplitudes. Lower plots show variables $x_1 - x_2$ in red, z in green and m in purple evolving for the same simulation for the region left hippocampus anterior. A) Empirical SEEG recording plot of a stimulation-induced seizure. Stimulation was applied at amplitude 1 mA using channels TB'1 [+], and TB'2[-], at frequency 50 Hz, pulse width 1 ms and duration 4 s. Reconstructed SEEG electrodes are shown on the left and stimulation location is plotted in red. B) Upper plot, synthetic SEEG time series of simulated brain activity with stimulation applied at 0.5 mA amplitude. All other stimulation parameters are identical to the empirical parameters. Here, a seizure is not induced after the stimulation is applied. Lower plot, the same simulated activity for the left hippocampus anterior, showing the variable m did not cross the seizure threshold, defined at 3.5 and corresponding variables staying in the normal state. C) Synthetic SEEG time series of the stimulation-induced seizure at 1 mA amplitude. Here, the same stimulation parameters as the ones applied empirically were used. Following the stimulation, a seizure is induced in the left hippocampus anterior, propagating later on to connected brain structures. Lower plot showing the variable m crossed the seizure threshold and the system is kicked to the seizure state. D) Synthetic SEEG time series plot of simulated brain activity with stimulation applied at 3 mA amplitude. Following the stimulation, a seizure is induced in the left hippocampus anterior and propagating later on to connected brain structures. Lower plot showing the variable m crossed the seizure threshold and the system is kicked to the seizure state.

Response Figure (9): Simulation examples with different stimulation location from virtualized patient 10. Double arrow indicates stimulation period. Seven channels are plotted in bipolar montage (out of 116 total bipolar channels). Vertical red lines indicate seizure onset and seizure offset. A) Upper plot, empirical SEEG recording of a stimulation-induced seizure. Stimulation was applied at amplitude 1 mA using channels TB'1 [+] and TB'2[-], at frequency 50 Hz, pulse width 1 ms and duration 4 s. Lower plot, corresponding simulated time series of a stimulation-induced seizure. B) Simulated time series of stimulation applied by electrodes located within 1 cm distance from the empirical stimulation location (TB'1-2). C) Simulated time series of stimulation applied by electrodes located between 1 and 2 cm distance from the empirical stimulation location. D) Simulated time series of stimulation applied by electrodes located between 2 and 3 cm distance from the empirical stimulation location. E) Simulated time series of stimulation applied by electrodes located more than 3 cm away from the empirical stimulation location.

Reviewer Comment 3.3 — Compare the amplitude and distribution of the EZN activation under SEEG and TI stimulations.

Reply:

Thank you so much for your valuable suggestion. Following your suggestion, we have compared the stimulation effects of SEEG stimulation and TI stimulation in the revision. Our quantified measures here can be directly applied to help design and verify whether non-invasive stimulation achieves similar focality and effects as invasive stimulation. We presented the results in Response Fig. 10, and included one paragraph in the last part of the results section and another in the Application: Ongoing scientific studies in the section discussion.

In addition, we provided quantified measures to compare the effects of SEEG and TI based on activated brain activity on the source level. We defined these effects in terms of spatial distribution and amplitude. Spatial distribution was assessed using the activated area and the largest activated distance. The stimulation effects immediately after stimulation stops are shown in Supplementary Fig. 10. SEEG stimulation leads to a larger seizing area (Supplementary Fig. 10 **A,C**) but a similar maximal distance

among these areas (Supplementary Fig. 10 **B**). In contrast, TI stimulation induced a lower activation amplitude (Supplementary 10 **D**).

In the section discussion:

We also provided quantified measures to compare activated brain activity between SEEG and TI stimulation. We performed the high-resolution brain activity simulations at the source level, allowing us to calculate the activated brain activity. One immediate application is to help design and verify that non-invasive stimulation achieves similar focality and effects as invasive stimulation. For example, the optimization of multipolar TI stimulations [30] promises to be as effective as invasive deep brain stimulation for Parkinson's disease and epilepsy. Efficiency can be defined in terms of spatial distribution (focality) and stimulated amplitude.

Response Figure (10): Comparison of the activated stimulation effect immediately after stimulation stops between SEEG and TI stimulation. $t = 0$ represents the moment when stimulation stops. Blue indicates SEEG stimulation, while orange represents TI stimulation.” (A). We first identified the seizure onset and offset times for vertices that show seizure activity at the source level. Then, we calculated the activated area at a specific time by identifying vertices where this specific time falls between their seizure onset and offset. (B). The largest activated distance at a specific time is defined as the maximum distance between all activated vertices where this specific time falls between their seizure onset and offset. (C). Unlike A, here the activated vertices are defined based on whether they are active at a specific time point. (D). The mean amplitude of the top 10 vertices, ranked by their amplitudes. (E). The brain activity at the source level simulation under SEEG and TI stimulation at two example time points, 930 ms and 2470 ms, marked as dots in A-D. The clips are extracted from the same videos as Fig. 6 B and Fig. 7 B.

References

[1] Bod, R. *et al.* Synaptic alterations and neuronal firing in human epileptic neocortical excitatory networks. *Frontiers in Synaptic Neuroscience* **15** (2023).

- [2] Kumar, S. S. & Buckmaster, P. S. Hyperexcitability, interneurons, and loss of gabaergic synapses in entorhinal cortex in a model of temporal lobe epilepsy. *Journal of Neuroscience* **26**, 4613–4623 (2006). URL <https://www.jneurosci.org/content/26/17/4613>.
- [3] Trebuchon, A. *et al.* Electrical stimulation for seizure induction during seeg exploration: A useful predictor of postoperative seizure recurrence? *Journal of Neurology, Neurosurgery and Psychiatry* **92**, 22–26 (2021).
- [4] Kovac, S., Kahane, P. & Diehl, B. Seizures induced by direct electrical cortical stimulation—mechanisms and clinical considerations. *Clinical Neurophysiology* **127**, 31–39 (2016).
- [5] Frauscher, B. *et al.* Stimulation to probe, excite, and inhibit the epileptic brain. *Epilepsia* **64**, S49–S61 (2023). Epub 2023 May 18. PMID: 37194746; PMCID: PMC10654261.
- [6] Azilinson, M. *et al.* Brain sodium mri-derived priors support the estimation of epileptogenic zones using personalized model-based methods in epilepsy. *Network Neuroscience* 1–41 (2024).
- [7] Friston, K. J. Functional and effective connectivity: A review. *Brain Connectivity* **1**, 13–36 (2011).
- [8] Buckmaster, P. S., Zhang, G. F. & Yamawaki, R. Axon sprouting in a model of temporal lobe epilepsy creates a predominantly excitatory feedback circuit. *The Journal of Neuroscience* **22**, 6650–6658 (2002).
- [9] Esclapez, M., Hirsch, J. C., Ben-Ari, Y. & Bernard, C. Newly formed excitatory pathways provide a substrate for hyperexcitability in experimental temporal lobe epilepsy. *The Journal of Comparative Neurology* **408**, 449–460 (1999).
- [10] Bernard, C. in *Alterations in synaptic function in epilepsy* 4th edn, (eds Noebels, J., Avoli, M., Rogawski, M. *et al.*) *Jasper’s Basic Mechanisms of the Epilepsies* (National Center for Biotechnology Information (US), Bethesda, MD, 2012). URL <https://www.ncbi.nlm.nih.gov/books/NBK98161/>.
- [11] Wendling, F., Bartolomei, F., Bellanger, J. J. & Chauvel, P. Interpretation of interdependencies in epileptic signals using a macroscopic physiological model of the eeg. *Clinical Neurophysiology* **112**, 1201–1218 (2001).
- [12] Depannemaecker, D., Ezzati, A., Wang, H., Jirsa, V. & Bernard, C. From phenomenological to biophysical models of seizures. *Neurobiology of Disease* 106131 (2023).
- [13] Palombo, M., Ligneul, C., Hernandez-Garzon, E. & Valette, J. Can we detect the effect of spines and leaflets on the diffusion of brain intracellular metabolites? *NeuroImage* **182**, 283–293 (2018). URL <https://www.sciencedirect.com/science/article/pii/S1053811917303932>. Microstructural Imaging.
- [14] Wang, H. E. *et al.* Delineating epileptogenic networks using brain imaging data and personalized modeling in drug-resistant epilepsy. *Science Translational Medicine* **15** (2023).
- [15] Jirsa, V. *et al.* Personalised virtual brain models in epilepsy. *The Lancet Neurology* (2023).

- [16] Ségonne, F. *et al.* A hybrid approach to the skull stripping problem in mri. *NeuroImage* **22**, 1060–1075 (2004). URL <https://www.sciencedirect.com/science/article/pii/S1053811904001880>.
- [17] Fischl, B. *et al.* Sequence-independent segmentation of magnetic resonance images. *NeuroImage* **23**, S69–S84 (2004). URL <https://www.sciencedirect.com/science/article/pii/S1053811904003817>. Mathematics in Brain Imaging.
- [18] Sled, J., Zijdenbos, A. & Evans, A. A nonparametric method for automatic correction of intensity nonuniformity in mri data. *IEEE Transactions on Medical Imaging* **17**, 87–97 (1998).
- [19] Segonne, F., Pacheco, J. & Fischl, B. Geometrically accurate topology-correction of cortical surfaces using nonseparating loops. *IEEE Transactions on Medical Imaging* **26**, 518–529 (2007).
- [20] Fischl, B., Sereno, M. I. & Dale, A. M. Cortical surface-based analysis: Ii: inflation, flattening, and a surface-based coordinate system. *Neuroimage* **9**, 195–207 (1999).
- [21] Destrieux, C., Fischl, B., Dale, A. & Halgren, E. Automatic parcellation of human cortical gyri and sulci using standard anatomical nomenclature. *NeuroImage* **53**, 1–15 (2010). URL <https://www.sciencedirect.com/science/article/pii/S1053811910008542>.
- [22] Wang, H. E. *et al.* Vep atlas: An anatomic and functional human brain atlas dedicated to epilepsy patients. *Journal of Neuroscience Methods* **348**, 108983 (2021). URL <https://linkinghub.elsevier.com/retrieve/pii/S0165027020304064>.
- [23] Desikan, R. S. *et al.* An automated labeling system for subdividing the human cerebral cortex on mri scans into gyral based regions of interest. *NeuroImage* **31**, 968–980 (2006). URL <http://linkinghub.elsevier.com/retrieve/pii/S1053811906000437>.
- [24] Zhao, Y. *et al.* The brain structure, immunometabolic and genetic mechanisms underlying the association between lifestyle and depression. *Nature Mental Health* **1**, 736–750 (2023).
- [25] Schaefer, A. *et al.* Local-global parcellation of the human cerebral cortex from intrinsic functional connectivity mri. *Cerebral Cortex* **28**, 3095–3114 (2018).
- [26] Dollomaja, B. *et al.* Virtual epilepsy patient cohort: generation and evaluation. *medRxiv* (2024). URL <https://www.medrxiv.org/content/early/2024/10/10/2024.10.02.24314607>.
- [27] Wang, H. E. *et al.* Virtual brain twins: from basic neuroscience to clinical use. *National Science Review* **11** (2024).
- [28] Hashemi, M. *et al.* On the influence of prior information evaluated by fully bayesian criteria in a personalized whole-brain model of epilepsy spread. *PLOS Computational Biology* **17**, e1009129 (2021). URL <https://dx.plos.org/10.1371/journal.pcbi.1009129>.
- [29] Lemaréchal, J.-D. *et al.* Effects of the spatial resolution of the virtual epileptic patient on the identification of epileptogenic networks. *Imaging Neuroscience* (2024).
- [30] Zhu, X. *et al.* Multi-point temporal interference stimulation by using each electrode to carry different frequency currents. *IEEE Access* **7**, 168839–168848 (2019).